



# Interaction between marine and terrestrial biogenic volatile organic compounds: Non-linear effect on secondary organic aerosol formation

Xiaowen Chen, Lin Du[*], Zhaomin Yang, Shan Zhang, Narcisse Tsona Tchinda, Jianlong Li, Kun Li[*]

Environment Research Institute, Shandong University, Qingdao, 266237, China

*Correspondence to*: Lin Du (lindu@sdu.edu.cn) and Kun Li (kun.li@sdu.edu.cn)

**Abstract.** Biogenic volatile organic compounds (BVOCs) are the largest source of secondary organic aerosols (SOA) globally. However, the complex interactions between marine and terrestrial BVOCs remain unclear, inhibiting our in-depth understanding of the SOA formation in the coastal areas and its environmental impacts. Here, we performed smog chamber experiments with mixed α-pinene (a typical monoterpene) and dimethyl sulfide (DMS, a typical marine emission BVOC) to

investigate their possible interactions and subsequent SOA formation. It is found that DMS has a non-linear effect on SOA generation: the mass concentration and yield of SOA show an increasing and then decreasing trend with the increase of the initial concentration of DMS. The increasing trend can be attributed to OH regeneration together with acid-catalyzed heterogeneous reactions by the oxidation of DMS, while the decreasing trend is explained by the high OH reactivity that inhibits the formation of low volatility products. The results from infrared spectra and mass spectra together reveal the

contribution of sulfur-containing molecules in the mixed system. Moreover, the mass spectra results indicate that acidic products generated by DMS photooxidation enhance the O:C ratio, while organosulfates are produced to contribute to the formation of mixed SOA. In addition, the trends in relative abundance of highly oxygenated organic molecules (HOMs) with $C_8$-$C_{10}$ multiple functional groups in different mixed systems agree well with the turning point of the SOA yield. The findings of this study have significant implications for understanding binary or more complex systems in the atmosphere in the coastal

areas.

## 1 Introduction

Secondary organic aerosol (SOA), a major component of atmospheric aerosols, has significant impacts on atmospheric chemistry, global climate, and human health (Ziemann and Atkinson, 2012; Bilsback et al., 2023; Lv et al., 2023; Liu et al., 2023). SOA is generated from the gas-phase oxidation of volatile organic compounds (VOCs), gas-particle partitioning and

particle-phase reactions (Chen et al., 2021; Iyer et al., 2021; Shen et al., 2021). A variety of organic gases are present in the atmosphere, including biogenic volatile organic compounds (BVOCs) and anthropogenic volatile organic compounds (AVOCs) (Bauwens et al., 2022; Vettikkat et al., 2023; Ziemann and Atkinson, 2012; Kirkby et al., 2023). On a global scale, BVOCs are the most important source of atmospheric organic gases and account for more than 90% of the total global organic gases



(Ma et al., 2022; Lun et al., 2020). BVOCs contain a variety of biogenic organic gases emitted by plants (e.g., isoprene,

monoterpenes, sesquiterpenes), marine emissions (e.g., dimethyl sulfide (DMS)) (Lun et al., 2020; Yu and Li, 2021),and so

on. As an important monoterpene, $\alpha$-pinene accounts for about 70% of BVOC emissions and has been also widely studied for

its ability to produce very low volatile oxidation products with high SOA yields (Czoschke et al., 2003; Yu and Li, 2021;

Kenseth et al., 2023).

The ambient atmosphere is a complex mixture of different organic gases, which can interact and mix to form SOA through

the same emission source or in the atmospheric transport (Voliotis et al., 2022; Malik et al., 2018; Luo et al., 2015).The

oxidation of organic gases with other VOCs produces more SOA with monoterpenes or sesquiterpenes in the mixed systems

(Vivanco et al., 2011; Vivanco et al., 2013; Emanuelsson et al., 2013). Most of the monoterpenes and sesquiterpenes contain

endocyclic carbon-carbon double bonds. Special molecular structures can enhance the reactivity of mixed systems and

accelerate the rate of oxidant consumption, while making the products tend to polymerize and promoting the particle nucleation

(Li et al., 2021; Ylisirniö et al., 2020; Yang et al., 2019; Salvador et al., 2020; Vizenor and Asa-Awuku, 2018; Dada et al.,

2023). In contrast, the inhibitory effect of the mixing of organic gases was first suggested in the mixed oxidation of isoprene

and $\alpha$-pinene by Mcfiggans et al. (2019) and was attributed to a combination mechanism of both oxidant scavenging and

product scavenging. Recent studies also supported this conclusion (Heinritzi et al., 2020; Li et al., 2022; Kari et al., 2019).

Mixing of organic gases has diverse impacts on the formation of SOA in different systems. Therefore, it is necessary to further

investigate the SOA formation and possible interactions in more mixed systems.

DMS is a major reactive gas in the marine boundary layer (Yu and Li, 2021). The estimated global average annual marine

emission of DMS is about 28 TgS (Lana et al., 2011), accounting for more than 90% of total oceanic sulfide (Shen et al., 2018).

DMS can be oxidized to methanesulfonic acid (MSA) and sulfuric acid ($H_2SO_4$), contributing significantly to oceanic new

particle generation, particle growth, and atmospheric chemical processes (Arquero et al., 2017; Fung et al., 2022). Air

pollutants emitted from land could transport to the offshore region (Tsai et al., 2012), thereby affecting the oxidation ability of

the marine boundary layer and the concentration of VOCs, and even altering the distribution of marine organic aerosols in

offshore waters (Brooks and Thornton, 2018; Wu et al., 2020). Likewise, the oceanic release of DMS and the formation of

sulfate also contribute to the formation of acid rain in coastal cities, and the derived MSA and organic sulfate (OS) contribute

significantly to the organic sulfide in the atmosphere (Huang et al., 2015). DMS can transport to coastal areas and subsequently

interact with other VOCs. Therefore, the mixing of DMS and other VOCs is significant in coastal areas and marine atmosphere.

However, the studies on mixing DMS with other VOCs are rare. To our knowledge, the only previous study about this was the

mixing of DMS and isoprene (Chen and Jang, 2012), and it demonstrated that the presence of DMS increased the SOA yield

of isoprene by forming $H_2SO_4$. The possible interactions between DMS and monoterpenes that may impact SOA formation

and composition, however, remain unexplored. In particular, the SOA formation processes from monoterpenes (e.g., $\alpha$-pinene)

are significantly influenced by acidity and consequently, acids (such as $H_2SO_4$ and MSA) generated by the oxidation of DMS



may affect SOA formation in the mixture system (Czoschke et al., 2003; Eddingsaas et al., 2012; Han et al., 2016; Gao et al., 2004; Deng et al., 2021).

In this study, smog chamber experiments were performed to simulate the SOA formation processes from mixed BVOC systems in marine atmosphere and coastal areas. The chemical composition and functional group information of SOA formed
from the oxidation of single VOC and the VOCs mixture were compared to get insights into the processes that influence SOA formation. The results here are helpful to improve our understanding of the interactions between two typical BVOCs (i.e., DMS and monoterpenes) that could impact SOA formation and composition.

## 2 Experimental methods

### 2.1 Smog chamber experiments

All experiments were performed in a 1.5 m³ indoor smog chamber to simulate the formation of SOA from the photooxidation of $\alpha$-pinene and DMS. The initial experimental conditions are summarized in Table 1. All experiments were conducted at temperature of 299 ± 1 K and relative humidity (RH) of 30–40%. These experiments were divided into three parts, namely individual $\alpha$-pinene photooxidation (Exp. A-1 to Exp. A-2), individual DMS photooxidation (Exp. D-1 to Exp. D-3) and mixed photooxidation of $\alpha$-pinene and DMS (Exp. AD-1 to Exp. AD-13). Black light lamps (F40BLB, GE) with center emission
wavelength of ~365 nm were used to initiate the photochemistry. The photolysis rate of $NO_2$ ($J_{NO2}$) was determined to be 0.26 min⁻¹ inside the reactor. The temperature and RH in the reactor were monitored by a hygro-thermometer (HM42PROBE, VAISALA, Finland). Prior to each experiment, the reactor was cleaned with dry, clean, and continuous zero air to ensure that $NO_x$, $SO_2$ and $O_3$ concentrations in the reactor were below 1 ppb and the background particle number concentration was below 10 cm⁻³.

A certain known amount of $\alpha$-pinene (Aladdin, 98%) and/or DMS (Macklin, 99.7%) was passed through a syringe into a Teflon tube and injected into the reactor with zero air. NO was introduced into the chamber from a gas cylinder (510 ppm in $N_2$, Qingdao Deyi Gas Company). OH radicals were generated from the photolysis of hydrogen peroxide ($H_2O_2$) (Aladdin, 30%). After all the reactants were introduced into the chamber, the reactor was stabilized in the dark for 20 min to ensure homogeneous mixing of the reactants. Then, the black lights were turned on to initiate photooxidation.

Table 1 Initial experimental conditions of the chamber experiments.

| Exp. No. | $[\alpha\text{-pinene}]_0$[a] ppb | $[DMS]_0$[a] ppb | $[NO]_0$[a] ppb | $[NO_x]_0$[a] ppb | T K | RH % |
|---|---|---|---|---|---|---|
| | individual $\alpha$-pinene | | | | | |
| A-1 | 308 | 0 | 195 | 206 | 300 | 37 |
| A-2* | 285 | 0 | 201 | 204 | 298 | 32 |



| Exp. No. | $[\alpha\text{-pinene}]_0{}^a$ ppb | $[DMS]_0{}^a$ ppb | $[NO]_0{}^a$ ppb | $[NO_x]_0{}^a$ ppb | T K | RH % |
|---|---|---|---|---|---|---|
| **individual DMS** | | | | | | |
| D-1 | 0 | 184 | 189 | 192 | 299 | 35 |
| D-2* | 0 | 290 | 206 | 211 | 299 | 34 |
| D-3* | 0 | 600 | 207 | 212 | 299 | 32 |
| **mix $\alpha$-pinene and DMS** | | | | | | |
| AD-1 | 312 | 140 | 208 | 211 | 299 | 33 |
| AD-2 | 321 | 197 | 190 | 195 | 299 | 31 |
| AD-3 | 305 | 301 | 203 | 212 | 299 | 34 |
| AD-4 | 291 | 372 | 183 | 203 | 300 | 40 |
| AD-5 | 308 | 441 | 193 | 202 | 298 | 34 |
| AD-6 | 317 | 536 | 191 | 203 | 299 | 33 |
| AD-7 | 282 | 639 | 196 | 206 | 299 | 33 |
| AD-8 | 306 | 613 | 189 | 193 | 298 | 33 |
| AD-9 | 295 | 687 | 183 | 191 | 299 | 32 |
| AD-10* | 319 | 251 | 197 | 200 | 298 | 36 |
| AD-11* | 314 | 401 | 184 | 194 | 298 | 33 |
| AD-12* | 332 | 646 | 198 | 199 | 298 | 37 |
| AD-13* | 300 | 614 | 193 | 197 | 299 | 32 |

[a] Initial concentrations of $\alpha$-pinene, DMS , NO and $NO_x$.

* For off-line analysis of SOA.

## 2.2 Online determination of gases and particles

A series of instruments were used to determine the gas-phase and particle-phase substances. The concentrations of $NO_x$, $O_3$ and $SO_2$ were monitored continuously and online by $NO$-$NO_2$-$NO_x$ analyzer (Model 42i, Thermo Scientific, USA), $O_3$ analyzer (Model 49i, Thermo Scientific, USA) and $SO_2$ analyzer (Model43i-TLE, Thermo Electron Corporation, USA), respectively. The time resolution of these analyzers was 10 s, and the measurement uncertainty was less than ±1%. $\alpha$-Pinene and DMS concentrations were measured with a gas chromatography-flame ionization detector (GC-FID, 7890B, Agilent Technologies, USA) with a DB-624 capillary column (30 m × 0.32 mm, 1.8 mm film thickness). The temperature for GC analysis was programmed to increase from 80 to 200 °C at 20 °C min$^{-1}$ rate. The number concentration, size distribution and volume concentration of the particles were measured with a scanning mobility particle sizer (SMPS, model 3938, TSI Inc., USA), which consists of an electrostatic classifier (TSI 3082), a differential mobility analyzer (TSI 3081) and a condensation particle



counter (TSI 3776). The aerosol flow rate and sheath flow rate of SMPS were 0.3 L min$^{-1}$ and 3 L min$^{-1}$, respectively. The particle volume concentration measured by SMPS was multiplied by an assumed aerosol density of 1.2 g cm$^{-3}$ to obtain the mass concentration of SOA (Turpin and Lim, 2001; Yu et al., 2008). The particle concentration was affected by the adsorption of particulate matter onto the smog chamber walls during the reaction. The wall loss rate constants of NO$_x$, O$_3$, SO$_2$, $\alpha$-pinene,

and DMS inside the chamber were $1.18 \times 10^{-6}$, $2.20 \times 10^{-6}$, $2.88 \times 10^{-6}$, $3.16 \times 10^{-6}$ and $7.42 \times 10^{-6}$ s$^{-1}$, respectively, indicating negligible losses of these gases over the course of the experiments. All the SOA mass concentrations were also corrected considering the wall loss of particles using the relationship between the deposition rate constant of a particle and its diameter, and the details are described in Sect. S1in the supplement.

## 2.3 Offline analysis of particles

Particle samples were collected on 47 mm PTFE filters (0.22 mm pore size, Tianjin Jinteng Experimental Equipment). The collected filters were wrapped with aluminum foil and stored in a -20 °C freezer prior to analysis. Details of the filter handling and extraction protocols can be found in Sect. S2 in the supplement. The chemical composition of SOA was determined by ultra performance liquid chromatography (UPLC, UltiMate 3000, Thermo Scientific) coupled with quadrupole time-of-flight mass spectrometry (Q-TOFMS, Bruker Impact HD). Electrospray ionization (ESI) was operated in the negative ion mode

using a 4 kV capillary voltage. For the analysis using UPLC/ESI-Q-TOFMS, the mobile phase was ultra-pure water containing 0.1% formic acid in phase A and methanol containing 0.1% formic acid in phase B. The elution flow rate was 0.2 mL min$^{-1}$. Aliquot of 5 μL of the sample solution was subjected to gradient elution in a column (Atlantis T3, 100 Å size, 3 μm particle size, 2.1 mm ID × 150 mm length, Waters, USA) using a gradient elution. The B phase was first held at a 3% gradient for 3 min, then it increased linearly from 3% to 50% in 25 min, followed by an increase to 90% in 18 min, and it finally dropped

back to 3% in 5 min and held for 12 min. The detection mode in the mass spectrometry parameters was a first level full scan (*m/z* 50–1500). Data were acquired and analyzed by Bruker Compass Data Analysis Version 4.2 software. Particulate samples were blank-corrected.

A low-pressure impactor (DLPI$^+$, DeKati Ltd) was used to collect the aerosol particles on the aluminum foil substrate. Functional groups and bond information of particle collected were analyzed using attenuated total reflectance coupled to the

Fourier transform infrared spectroscopy (ATR-FTIR, Vertex 70, Bruker, Germany). For the measurement, the sampled aluminum foil was placed on a clean ATR crystal, and the background was deducted using a blank aluminum foil. The scanning wavenumber range was set to 4000–600 cm$^{-1}$, the number of scans was 64, and the resolution was 4 cm$^{-1}$. Data processing of the acquired IR spectra was performed using the rubberband method in the OPUS software package.

During the experiments, sulfate and nitrate may have been generated in the particle phase. For this reason, the collected

aerosol particles were subjected to ion chromatography. Before the measurement, particle samples were collected on 47 mm PTFE filters (0.22 mm pore size, Tianjin Jinteng Experimental Equipment). The sampling filter was dissolved with 4 mL of ultrapure water, sonicated in a water bath for 30 min, and filtered through a 0.22 μm aqueous needle filter. The samples were determined by ion chromatography (Dionex ICS-600, Thermo Fisher, USA) with an injection volume of 250 μL. The anions



were determined using a 20 mM potassium hydroxide solution as a washout, flowed through an anion-protected column
(Dionex AG18, $4 \times 50$ mm) at a flow rate of 1 mL min$^{-1}$, and separated by an anion column (Dionex IonPacTM AS19, $4 \times 250$ mm). The cationic eluent was 20 mM MSA, the guard column was a Dionex AG18, $4 \times 50$ mm, and the column was a Dionex IonPac CS12A, $4 \times 250$ mm.

## 2.4 Gas-phase chemistry modelling

A zero-dimensional (0-D) box model with the Master Chemical Mechanism (MCM) v3.3.1 (Jenkin et al., 2015), i.e.,
Framework for 0-D Atmospheric Modeling (F0AM) (Wolfe et al., 2016), was used to simulate the time series of $\alpha$-pinene, DMS and OH during the chamber experiments. For these simulations, the environmental parameters (temperature, humidity, etc.) and the initialized concentrations of the reactants in the model were consistent with each experiment. We ran the SOA model using a MATLAB-based package. The main chemical mechanism (MCM v3.3.1) is a near-explicit mechanism with more detailed reactions (http://mcm.york.ac.uk/) (Jenkin et al., 2015). Therefore, we utilized the gas-phase chemical
mechanism for DMS and $\alpha$-pinene from the MCM v3.3.1 website for our analysis.

## 3 Results and discussion

### 3.1 Secondary particle formation

Tables 2 and 3 show the experimental results of gas-phase and particle-phase components in photooxidation of DMS and $\alpha$-pinene systems. The photooxidation of DMS could generate gas-phase products including $SO_2$, dimethyl sulfoxide (DMSO)
and dimethyl sulfone ($DMSO_2$), and particle-phase products such as $H_2SO_4$, MSA and methyl sulfite (MSIA) (Abbatt et al., 2019; Sciare et al., 1998; Jiang et al., 2022; Ye et al., 2022). In Exp. D-1, DMS is consumed completely after 250 min of light exposure (Fig. 1), generating 67 ppb $SO_2$, 50.8 µg m$^{-3}$ particle-phase $H_2SO_4$ and 32.8 µg m$^{-3}$ particle-phase MSA (Table 3). $H_2SO_4$ and MSA together account for 47% of the particle mass concentration. In a previous study, the mass concentrations of $H_2SO_4$ and MSA in the particle phase were 8.4 µg m$^{-3}$ and 48 µg m$^{-3}$, respectively, when DMS consumption in the reaction
was 71 ppb (Chen and Jang, 2012). The ratio of $H_2SO_4$ to MSA in this study (~1.5) is significantly higher than that in the previous work (~0.18), which attributed to the different experimental conditions of the two experimental systems, e.g., RH (32% in this study vs 10% in the previous study), DMS consumption (183 ppb vs 71 ppb), DMS product collection method (filters vs PILS), oxidant ($H_2O_2$ vs NO$_x$), etc.



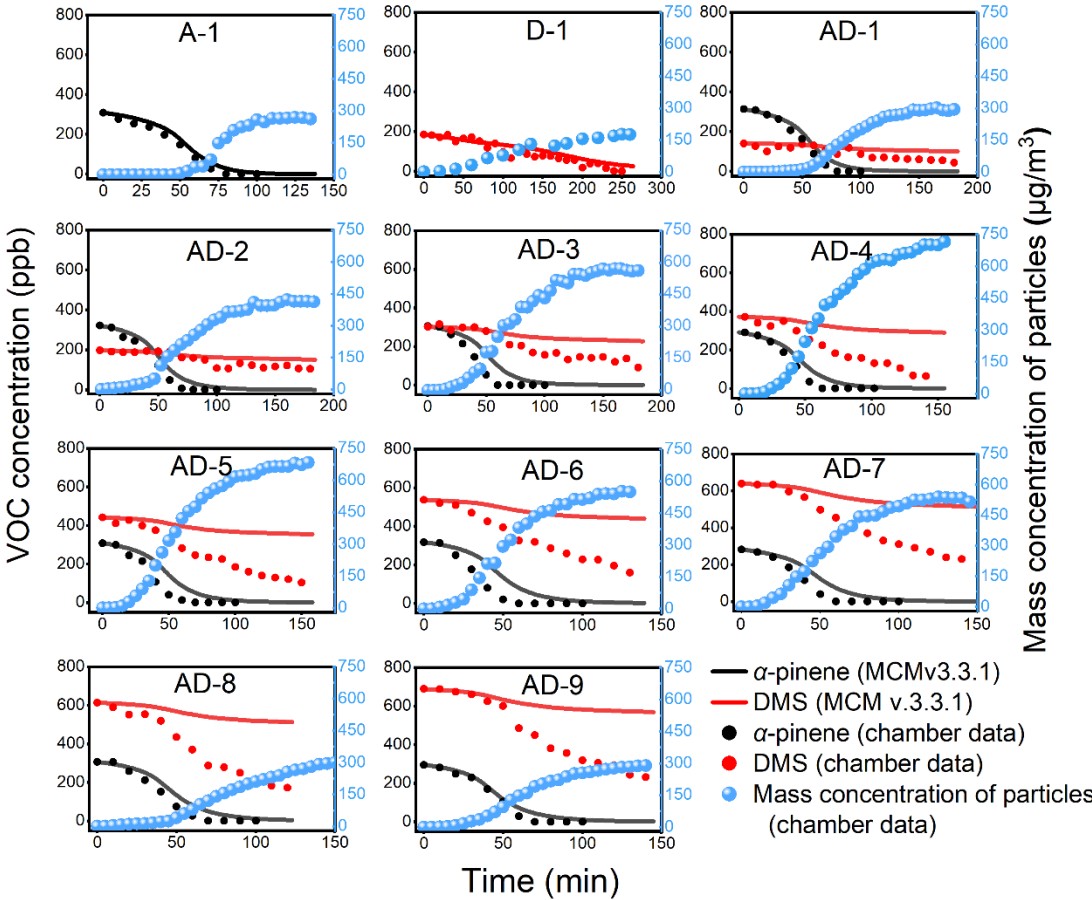

**Figure 1. Variation of precursors with reaction time. Red and black dots indicate the results of smog chamber experiments and the curves indicate the results of MCM simulations. Blue dots indicate mass concentration of particles in smog chamber.**

The particulate matter generated by DMS photooxidation mainly contains two types of components, $H_2SO_4$ and SOA. Hence, the individual and mixed experiments of DMS contain a certain amount of sulfur-containing inorganic components. As can be seen from Fig. 1, SOA particles are gradually produced after 50 min in the individual $\alpha$-pinene experiment, while in the individual DMS experiment, the particles are produced immediately after the start of the photooxidation. When DMS was added to the oxidation of $\alpha$-pinene, particles formation was advanced, and the total amount of particles is more than in Exp. A-1 as the photooxidation continued. This indicates that the oxidation products of DMS can promote the particles formation.

Table 2 Experimental results of gas-phase components in photooxidation of DMS and $\alpha$-pinene systems.

| Exp. No. | $\Delta[\alpha\text{-pinene}]^a$ ppb | $\Delta[\text{DMS}]^a$ ppb | $[NO]^b$ ppb | $[NO_x]^b$ ppb | $[SO_2]^b$ ppb | $[O_3]_{max}^c$ ppb |
|---|---|---|---|---|---|---|
| | | | individual $\alpha$-pinene | | | |
| A-1 | 308 | - | 0 | 87 | - | 21 |



| Exp. No. | $\Delta[\alpha\text{-pinene}]^a$ ppb | $\Delta[DMS]^a$ ppb | $[NO]^b$ ppb | $[NO_x]^b$ ppb | $[SO_2]^b$ ppb | $[O_3]_{max}^c$ ppb |
|---|---|---|---|---|---|---|
| A-2* | 285 | - | 0 | 64 | - | 30 |
| **individual DMS** | | | | | | |
| D-1 | - | 183 | 0 | 33 | 67 | 19 |
| D-2* | - | 276 | 0 | 35 | 101 | 25 |
| D-3* | - | 372 | 2 | 98 | 110 | 41 |
| **mix $\alpha$-pinene and DMS** | | | | | | |
| AD-1 | 312 | 83 | 0 | 79 | 31 | 25 |
| AD-2 | 321 | 87 | 0 | 74 | 35 | 25 |
| AD-3 | 305 | 181 | 0 | 55 | 69 | 23 |
| AD-4 | 291 | 307 | 0 | 42 | 87 | 18 |
| AD-5 | 308 | 338 | 0 | 50 | 111 | 25 |
| AD-6 | 317 | 359 | 0 | 67 | 97 | 24 |
| AD-7 | 282 | 384 | 0 | 58 | 124 | 27 |
| AD-8 | 306 | 440 | 1 | 78 | 160 | 45 |
| AD-9 | 295 | 457 | 0 | 69 | 154 | 51 |
| AD-10* | 319 | 219 | 0 | 61 | 75 | 25 |
| AD-11* | 314 | 330 | 0 | 73 | 109 | 22 |
| AD-12* | 332 | 447 | 0 | 74 | 166 | 26 |
| AD-13* | 300 | 406 | 0 | 58 | 136 | 29 |

[a] The consumption of $\alpha$-pinene and DMS when the particles were produced to the maximum mass concentration determined by SMPS.

[b] The concentration of NO, $NO_x$ and $SO_2$ when the particles are produced to the maximum mass concentration.

[c] The maximum concentration of $O_3$ production during light exposure.

* For off-line analysis of SOA.

Table 3 Experimental results of particle-phase components in photooxidation of DMS/$\alpha$-pinene/$NO_x$ systems.

| Exp. No. | [Total particles] $^a$ $\mu g\ m^{-3}$ | $[H_2SO_4]^b$ $\mu g\ m^{-3}$ | $[MSA]^b$ $\mu g\ m^{-3}$ | $[SOA_t]^c$ $\mu g\ m^{-3}$ | $Y_t^d$ |
|---|---|---|---|---|---|
| **individual $\alpha$-pinene** | | | | | |
| A-1 | 269.5 | - | - | 269.5 | 0.16±0.02 |



| Exp. No. | $\Delta[\alpha$-pinene$]^a$ ppb | $\Delta[DMS]^a$ ppb | $[NO]^b$ ppb | $[NO_x]^b$ ppb | $[SO_2]^b$ ppb | $[O_3]_{max}{}^c$ ppb |
|---|---|---|---|---|---|---|
| | | | **individual DMS** | | | |
| D-1 | 177.2 | 50.8 | 32.83 | 116.2 | 0.25±0.03 | |
| | | | **mix $\alpha$-pinene and DMS** | | | |
| AD-1 | 296.3 | 15.0 | 22.18 | 270.8 | 0.14±0.02 | |
| AD-2 | 422.3 | 15.7 | 22.48 | 400.1 | 0.20±0.02 | |
| AD-3 | 572.6 | 45.4 | 24.90 | 507.5 | 0.24±0.02 | |
| AD-4 | 714.4 | 55.4 | 50.52 | 607.7 | 0.25±0.03 | |
| AD-5 | 683.0 | 48.8 | 36.15 | 613.1 | 0.24±0.02 | |
| AD-6 | 551.5 | 35.3 | 8.47 | 504.2 | 0.19±0.02 | |
| AD-7 | 539.9 | 48.4 | 16.77 | 476.0 | 0.19±0.02 | |
| AD-8 | 364.4 | 68.7 | 0.05 | 237.1 | 0.08±0.01 | |
| AD-9 | 289.9 | 83.2 | 6.98 | 154.0 | 0.06±0.01 | |

[a] The mass concentration of particles generated by SMPS, corrected for particle wall loss, was calculated as a particle density of 1.2 g cm$^{-3}$.

[b] IC detection, particle-phase products generated by DMS photooxidation.

[c] The mass concentration of the total SOA produced by the photooxidation reaction is expressed as [Total particles]$_{after\text{-}correction}\times(1-[H_2SO_4]/[Total\ particles]_{before\text{-}correction})$.

[d] [SOA$_t$]/($\Delta[\alpha$-pinene$]+\Delta[DMS]$), as mixed yield. Error bars indicate SMPS instrument error of 10%.

## 3.2 Effect of DMS on the yield of SOA

The total amount of SOA produced from the oxidation of mixed VOCs can be represented as the sum of the contributions of the individual molecular constituents. The consumptions of $\alpha$-pinene and DMS in Exp. AD-3 are closest to those in Exp. A-1 and Exp. D-1. The combination of Exp. A-1 and Exp. D-1 can be used to predict the mass concentration of SOA generated by

Exp. AD-3. The calculation process is shown in Sect. S3 in the supplement. The measured and predicted SOA mass concentrations in Exp. AD-3 are 507.5 µg m$^{-3}$ and 380.9 µg m$^{-3}$, respectively, indicating that the mixing of DMS with $\alpha$-pinene promoted the production of SOA. DMS oxidation generates acidic products such as SO$_2$, H$_2$SO$_4$, and MSA (De Jonge et al., 2021; Czoschke et al., 2003). DMS also increases the number concentration of particles while slightly increasing the particle size (Fig. S4), which is consistent with the findings of Xu et al. (2021). H$_2$SO$_4$ can nucleate rapidly, which makes the particle

number concentration increase in the presence of DMS and contributes to the gas-particle partitioning of small molecule products. In addition, H$_2$SO$_4$ promotes the uptake of gas-phase organic species through acid-catalyzed non-homogeneous





reactions, which produce low volatile organic compounds (LVOCs) and increase particle mass concentration (Kulmala et al., 2004; Jang et al., 2004; Jang and Kamens, 2001).

Since the addition of DMS can promote the photooxidation of $\alpha$-pinene, we further investigated the effect of different concentration of DMS on the generation of SOA in the mixed systems. Figure 2a demonstrates the variation of mass concentration and yield of SOA in the mixed system (Exp. AD-1 to Exp. AD-9) with the ratio of precursor consumption. With the increase of $\Delta[\text{DMS}]/\Delta[\alpha\text{-pinene}]$ (from 0 to 1.55), the total SOA mass concentration increased from 269.5 μg m$^{-3}$ to 613.1 μg m$^{-3}$ and then decreased to 154.0 μg m$^{-3}$, whereas the variation characteristic of the yield was consistent. The highest SOA mass concentration and SOA yield both appeared at $\Delta[\text{DMS}]/\Delta[\alpha\text{-pinene}]$ ~1 (i.e., the turning point, Exp. AD-4 and Exp. AD-

5). In addition to acid catalysis, changes in the average OH concentration ([OH]$_{avg}$) in different experiments (Fig. 2b) also affect SOA production. [OH]$_{avg}$ in the mixed experiments were all higher than that in individual $\alpha$-pinene oxidation. With the increase of $\Delta[\text{DMS}]/\Delta[\alpha\text{-pinene}]$, [OH]$_{avg}$ showed a trend consistent with the mass concentration and yield of SOA.

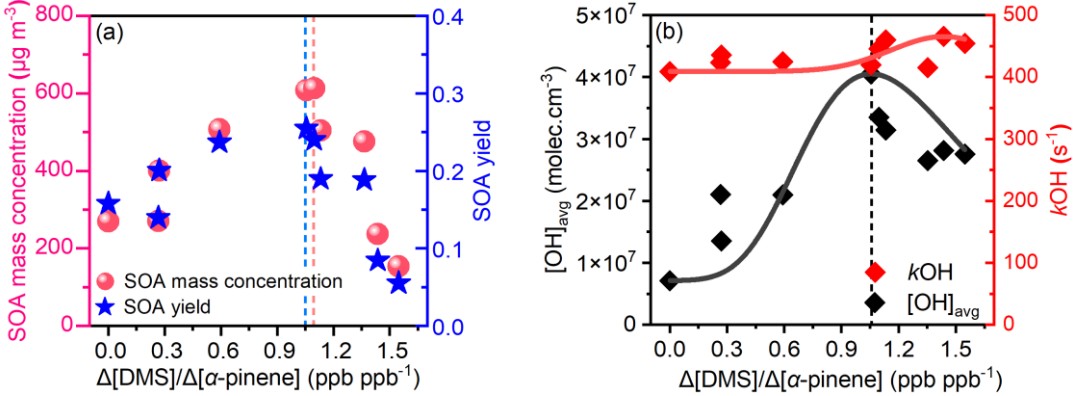

**Figure 2. Relationship between SOA mass concentration, yield (a) and average OH concentration ([OH]$_{avg}$), OH reactivity($k$OH) (b)**
**to VOCs consumption ratio. The solid lines show the fitting results.**

To more accurately reflect the [OH]$_{avg}$ of each experiment, we combined the MCM model to calculate the trends of OH concentrations in different experiments with time, as shown in Fig. S5. The maximum OH concentrations before the turning point from Exp. AD-1 to Exp. AD-4 are higher than those after the turning point from Exp. AD-5 to Exp. AD-9 at the end of experiments. Interestingly, the largest OH concentration formed in Exp. AD-4 during the time period when the OH

concentration was rising at the fastest rate from the local magnification graph, which is consistent with the average OH concentration reflected in Fig. 2b. The OH concentrations of other systems are also largely consistent. The OH concentration may increase in the mixed systems with increasing DMS concentration before the turning point. OH regeneration before the turning point could be attributed to the enhancement in SOA formation. Figure S8 shows the possible regeneration pathways of OH during DMS oxidation. DMS forms CH$_3$SCH$_2$O$_2$ radical through the hydrogen extraction channel, and this peroxyl

radical undergoes an isomerization process to form HOOCH$_2$SCHO product, accompanied by OH regeneration. It has been previously found that this isomerization channel resulted in a 43% increase in OH radical concentration by modeling or kinetic



calculations, and that the isomerization process of $CH_3SCH_2O_2$ radicals was significantly faster than the bimolecular reaction (Berndt et al., 2019; Wu et al., 2015).

The concentration of DMS in the mixed system increases significantly after the turning point, resulting in an increasing trend of total OH reactivity ($k$OH) (Fig. 2b). High $k$OH leads to a decrease in the utilization of OH participating in the multiple generation oxidation of VOCs, which in turn leads to a decrease in the concentration of $[OH]_{avg}$ in the reaction system. The occurrence of this situation results in the decrease of substances with relatively low volatility that require more oxidation steps (via OH) to significantly condense onto the particulate phase (Zhao et al., 2015; Yang et al., 2016). Although the acid catalysis of DMS still plays a certain role in promoting SOA formation, the increase of $k$OH leads to a decreasing trend of SOA yield after the turning point.

## 3.3 Effect of DMS on SOA composition

### 3.3.1 Comparison of SOA components for individual and mixed systems

To further determine the differences in the chemical composition of SOA in separate and mixed oxidation systems, we measured the SOA chemical composition using UPLC/ESI-Q-TOF-MS. Figure 3a shows the mass spectra of SOA in negative ion mode. SOA produced under both reaction conditions contains many common components. In the presence of DMS, the ion peak of $m/z$ 97 ($HSO_4^-$) shows a higher abundance. The presence of DMS contributes to $H_2SO_4$ formation, which influences the process of SOA formation through acid catalysis. Comparing the signal intensities of the mass spectral peaks in different $m/z$ ranges (Fig. 3b), it is found that the addition of DMS results in a decrease in the abundance of low molecular weight monomeric compounds ($m/z < 200$ and $200 \leq m/z \leq 299$) in SOA, while intensity of molecules with high molecular weight ($300 \leq m/z \leq 399$ and $m/z > 400$) is stronger in the mixed systems than in individual $\alpha$-pinene system. This further illustrates the presence of acid-catalyzed heterogeneous reactions in hybrid photooxidation, which transform low molecular weight compounds into substances with higher molecular weights.

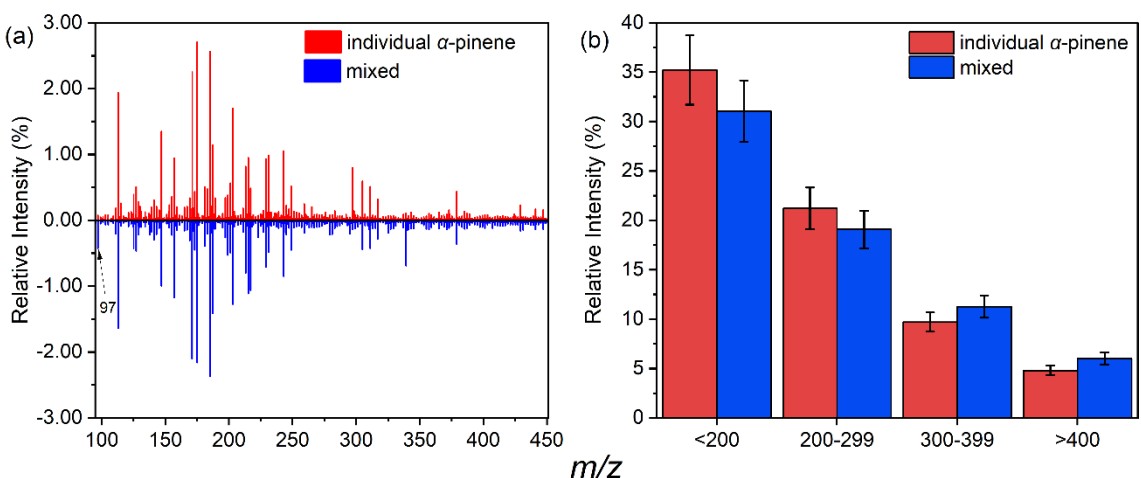



**Figure 3. Comparison of negative mode mass peak of SOA in the individual α-pinene and mixed systems. (a) Mass spectra of SOA with/without the presence of DMS. (b) Comparison of the relative intensities of mass spectrometry peaks with different *m/z* ratio ranges. Relative strength is the strength of a substance with a certain mass-to-charge ratio divided by the total strength of all substances.**

In order to characterize the changes in the elemental composition of particulate matters in different oxidation systems, we compared the Van Krevelen plot of the individual and mixed systems, as shown in Fig. S6 and Fig. S7. The H:C and O:C values of the particle-phase products were mainly distributed in the range of 0.5-2.2 and 0.2-1.2, in general agreement with the previous results of α-pinene SOA (Kourtchev et al., 2014; Thomsen et al., 2022). α-Pinene contains two rings with one double bond and has a double bond equivalents (DBE) of 3. During the photooxidation reaction, the six-membered ring is opened and the double bond is broken, but at the same time the C=O double bond is also formed. Isomerization and cleavage reactions further increase the production of monomer molecules of DBE around 3 for α-pinene photooxidation, while the appearance of molecules with larger DBE values may be associated with oligomerization molecules produced by the polymerization reaction (Laskin et al., 2010; Putman et al., 2012; Han and Jang, 2023).

The molecules in the mixed oxidation have a greater average O:C value than individual systems as shown in Fig. S6. The addition of DMS makes the molecular distribution tend to be in a higher range of O:C. These larger O:C molecules in the mixed system have low DBE values (Fig. S7). It can be seen from Fig. 4 that the molecules with higher O:C and lower DBE are attributed to sulfur-containing substances. These sulfur-containing compounds with low DBE values are typical for organosulfates (OSs) generated from monoterpenes (Surratt et al., 2008; Chan et al., 2011; Lin et al., 2012). As mentioned in Sect. 3.2, the oxidant level (OH concentration) is higher in the mixed oxidation than in individual α-pinene oxidation. The elevated oxidant level during the reaction helps to add more oxygen atoms to the product and enhances the degree of oxidation, resulting in a mixed system SOA with more high O:C compounds (Deng et al., 2021).

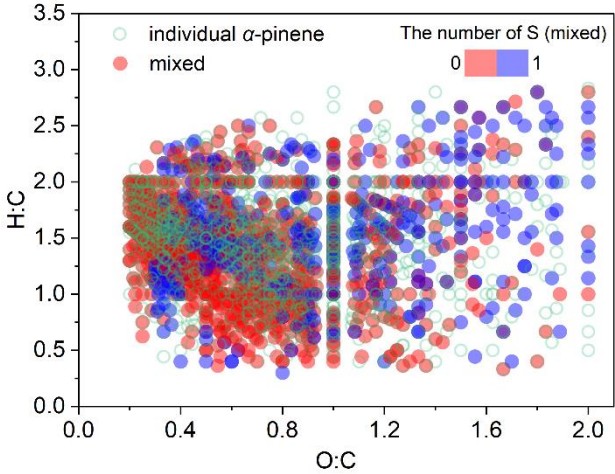

**Figure 4. Van Krevelen plots of compounds formed from α-pinene and mixed system. The color of the circle in mixed system refers to the number of S.**



Molecular Carbon Oxidation State ($OS_C$) is another way to characterize the degree of oxidation of complex molecules. Figure 5 demonstrates the variation of $OS_C$ with the number of carbon atoms for products in SOA in the separate and mixed

systems. Fragmentation products (C < 10), monomers (C = 10), and dimers (C > 10) were observed in both individual and mixed experiments. Compared to the individual $\alpha$-pinene oxidation system, the coexistence of DMS with $\alpha$-pinene resulted in higher $OS_C$ of S-containing molecules, indicating that the addition of DMS indeed altered the formation of these molecules.

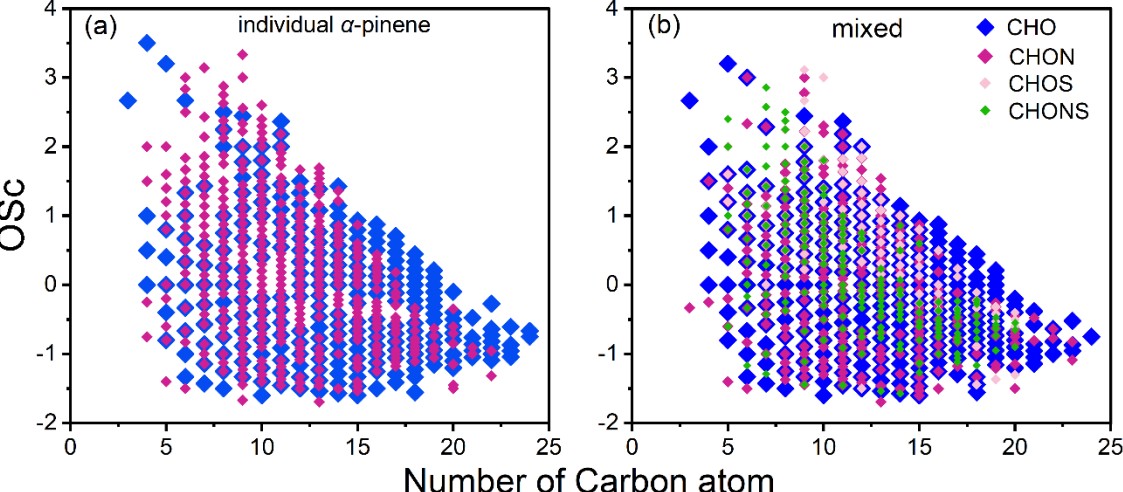

**Figure 5. The plots of $OS_C$ against carbon number of particulate organic molecules formed from individual $\alpha$-pinene and mixed**
**conditions.**

Figure 6 shows the volatility distribution of the particle phase products containing different elemental compositions. Overall, most compounds are distributed between the region of semi-volatile organic compounds (SVOC) to extremely low volatility organic compounds (ELVOC). The mixed systems photooxidation reaction is capable of generating a wide range of extremely low volatility products, and most of these molecules are CHOS- and CHONS-containing substances. Obviously, the

addition of DMS reduces the volatility distribution of high molecular weight molecules and the detected sulfur-containing molecules are mostly ELVOC, effectively reducing the product volatility. The low volatile OS is generated by particle phase reaction, thus providing evidence that $H_2SO_4$ promotes the occurrence of particle-phase reaction. On the other hand, the addition of DMS resulted in a wider O:C distribution of the products in the range of ELVOC and LVOC.



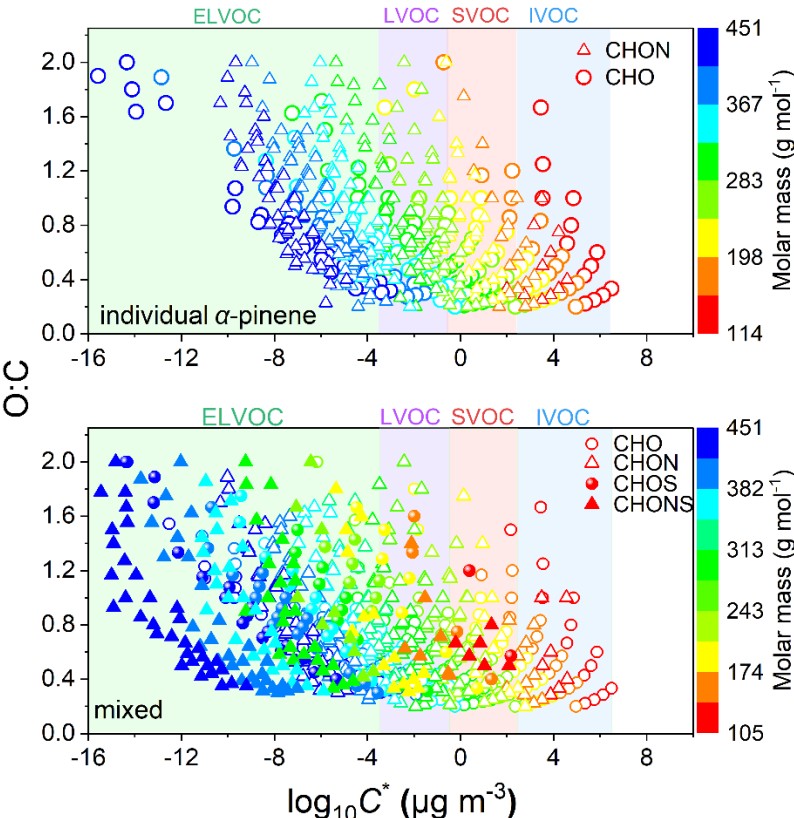

**Figure 6. Relationship between O:C and saturation concentration for molecules of different element types in individual α-pinene oxidation (a) and mixed oxidation (b).**

Table S2 shows the organic molecules identified by individual α-pinene oxidation and the mixed oxidation after the addition of DMS. It is indicated that some common components were generated in both individual and mixed systems, most of which are molecules containing CHO elements, and their relative intensity mostly differed insignificantly or irregularly.

The presence of DMS in the α-pinene system gives rise to sulfur-containing products (Fig. 7a). A total of six sulfur-containing compounds were distinguished in the particulate samples in the mixed system, which contributed to the particulate mass. The sulfur-containing compounds identified were determined to be organosulfates with oxygen atoms greater than 3 and the presence of an oxygen-to-carbon ratio greater than 1 or close to 1. Combined with Fig. 4, it can be found that there are OSs with O:C greater than 1 in the mixed system, which are defined as high oxidation state of OSs (HOOSs) that has been found previously in field observations (Mutzel et al., 2015). The formation of HOOSs has a correlation with HOMs, RO$_2$, and sulfate (Mutzel et al., 2015). The strength of these OSs varies in different systems. As shown in Fig. 7a, $C_{10}H_{16}N_2O_{10}S$ (MW 356), $C_{10}H_{15}NO_9S$ (MW 325), $C_{10}H_{17}NO_8S$ (MW 311), and $C_{10}H_{17}NO_7S$ (MW 295) had the highest relative intensity in the middle initial DMS concentrations experiments, i.e., lower or higher initial DMS concentrations inhibited their production, consistent with the trend of SOA yield. Combined with Table S2, DMS slightly generates the production of $C_6H_{10}O_8S$ (MW 242) and $C_5H_8O_8S$ (MW 227), which was attributed to the fact that the addition of DMS led to fragmentation pathway in the reaction.



The abundance of these two fragmented sulfur-containing molecules varied irregularly in different systems, which could not explain the emergence of the turning point.

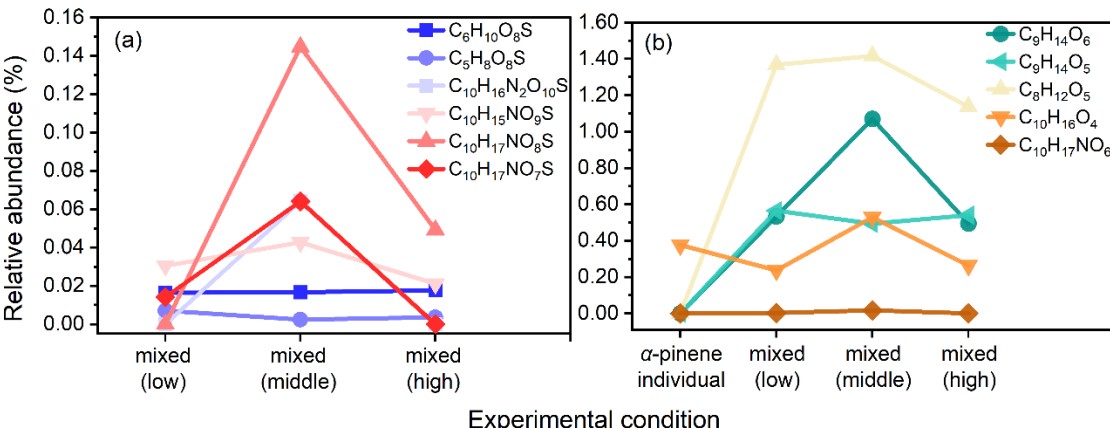

**Figure 7. Relative abundance of molecules identified in different reaction systems. (a) S-containing components in the mixed oxidation at different DMS concentrations. (b) CHO/CHON components recognized in individual and mixed oxidation. Details of the molecules are given in Table S2.**

The absence or presence of DMS coexisting in the $\alpha$-pinene photooxidation system can also impact the substances containing only CHO or CHON elements. After the addition of DMS to the reaction, some new molecules such as $C_9H_{14}O_6$ (MW 218), $C_9H_{14}O_5$ (MW 202), and $C_8H_{12}O_5$ (MW 188) with relatively high abundance were newly generated (Fig. 7b). We also notice that CHONS compounds (Fig. 7a) and almost all the molecules (Fig. 7b) contribute the most to the particle phase composition of SOA in the middle initial DMS concentrations experiments, which is a pattern consistent with the variation of SOA yield in different systems.

### 3.3.2 Effect of DMS on the mechanism of $\alpha$-pinene SOA generation

There are two pathways for the reaction of DMS with OH radicals: one is the H abstraction from the $CH_3$ fragment of DMS by OH, and the other is the addition of OH radicals to sulfur atoms. At room temperature, H abstraction is the main reaction pathway, accounting for ~70% of the total reaction (Barnes et al., 2006). As shown in Fig. S8, this pathway first generates the $CH_3SCH_2$ radical, which can add to $O_2$ to generate the $CH_3SCH_2O_2$ radical, followed by reaction with $RO_2$ or $HO_2$ to generate peroxy hydroxyl or peroxy nitrate, or with NO, eventually generating $SO_2$, MSA and $H_2SO_4$.

Some CHON compounds were generated in different experiments. Table S2 illustrates some of the organic nitrates (ONs) products generated in individual and mixed experiments. Based on previous studies (Boyd et al., 2015; Draper et al., 2015; Kim et al., 2012), Fig. 8a shows the two paths of ONs that we identified. One is that $\alpha$-pinene is oxidized to form $RO_2$, which is further oxidized to form alkoxyl radicals. Alkoxyl radicals form aldehydes by H-shift and oxygen addition. Hydrogen atom on the carbon chain of aldehyde is easily transferred, and then form nitrogen-containing carboxylate products by the addition of oxygen, i.e., $C_{10}H_{15}NO_6$ (MW 245). Due to the presence of $HO_2$, these first-generational ONs can be further oxidized to form multifunctional ONs containing carbonyl, carboxyl, and hydroxyl groups. This pathway occurs only in the individual





oxidation of $\alpha$-pinene. The second is that OH oxidizes $\alpha$-pinene to form $RO_2$, and it undergoes multiple isomerization and H-shift by the addition of oxygen to generate other $RO_2$, which can then form unsaturated ONs containing hydroxyl and hydroperoxy groups in mixed oxidation, i.e., $C_{10}H_{17}NO_6$ (MW 247). The presence of these ONs is consistent with IR spectra results, suggesting that one of the main ways in which $NO_x$ affects SOA formation is through multiple generational oxidation.

The production of the low volatility products can increase the particulate SOA yield.

In addition, Fig. 8 also demonstrates the possible structure of a high molecular weight oligomers generated in the individual $\alpha$-pinene experiments: $C_{20}H_{33}NO_8$ (MW 415). It is speculated that $RO_2$ tends more towards isomerization processes such as autoxidation compared to fragmentation reaction. This pathway increases the possibility of oligomerization of $RO_2$ with $RO_2$ and $RO_2$ with $HO_2$ in individual $\alpha$-pinene oxidation.

**Figure 8. Proposed formation mechanisms for organic nitrates in SOA. They are tentatively identified as organic nitrates. Red and blue in the box refer to the product identified by $\alpha$-pinene SOA and mixed SOA, respectively.**

In the compositional analysis of SOA generated by photooxidation of DMS-$\alpha$-pinene, a number of OSs were generated. Based on previous studies (Aschmann et al., 2002; Surratt et al., 2008; Surratt et al., 2007; Gao et al., 2006), we tentatively

proposed their formation pathways. Figure 9 shows the reaction pathway for $C_5$ organosulfate generation. The oxidation



product, pinonic acid reacts with OH under high-NO conditions to form 2/3-hydroxyglutaric acid (3-Hydroxyglutaric acid as an example), followed by sulfation to form $C_5H_8O_8S$ (MW = 228) (Surratt et al., 2008).

**Figure 9. Reaction pathway of C₅ organosulfate generation.**

Figure 10 presents the reaction pathways for the generation of three $C_{10}$ OSs. In the first pathway, $\alpha$-pinene undergoes OH addition, isomerization, and oxidation to form an organic nitrate followed by sulfation to form $C_{10}H_{17}NO_8S$ (MW 311). Furthermore, isomerization reaction of the first-generation hydroxyalkoxy radicals generated by the oxidation of $\alpha$-pinene and the organic nitrates generated in this pathway contains two OH groups.

     The second pathway involves the formation of hydroxyalkyl radicals by photooxidation of $\alpha$-pinene in the presence of
$NO_x$. This radical isomerizes the radical by H-abstraction and then immediately reacts with $O_2$ to form a hydroxyperoxy radical. Similarly, Surratt et al. (2008) found that $NO_3$ radical-driven oxidation of $\alpha$-pinene in the presence of highly acidic seed aerosols can also form hydroxyperoxy radicals, which then undergoes sulfation with $RO_2$ to form nitrogen-containing OSs. In our study, $C_{10}H_{15}NO_9S$ (MW 325) may be generated by the interaction of hydroxyorganic nitrate formed from $\alpha$-pinene by $NO_3$ oxidation with $H_2SO_4$ under the condition of no additional addition of acid sulfate. This is consistent with the inference
that $NO_2$ and $SO_2$ affect the production of SOA through the formation of $NO_3$ and $H_2SO_4$, respectively.

     The third pathway is the formation of $C_{10}H_{17}NO_7S$ (MW 295) from hydroxyperoxyl radicals generated by the oxidation of $\alpha$-pinene in the presence of NO and $H_2SO_4$. The third pathway usually occurs in the particulate phase. $C_{10}H_{17}NO_7S$ is formed by heterogeneous reactions of $SO_2$ with intermediates (Wang et al., 2020).





**Figure 10. Reaction pathway of $C_{10}$ organosulfate generation.**

The structures of other sulfur-containing organic compounds were also designated as shown in Fig. 11. Surratt et al. (2008) had identified the $C_{10}H_{16}N_2O_{10}S$ (MW 356) product and proposed that it would form in the presence of highly acidic seeds that first produces hydroperoxides containing two nitro groups, which further eliminate one water molecule by sulfation. Xu et al. (2020) also detected the isomer of $C_6H_{10}O_8S$ (MW 242), which was speculated to be a fragmentation product formed by further oxidation by ring structure breakage during the reaction.

**Figure 11. Structural formula of other identified organosulfates.**

Overall, the effects of DMS on the photooxidation of $\alpha$-pinene to SOA in the mixed systems with the coexistence of DMS are multiple. First, the $SO_2$ generated by DMS photooxidation, which reacts with OH in the gas-phase to generate $H_2SO_4$, plays



a central role in the generation of new particles and can facilitate the partitioning of gas-phase products into the particle phase, and the formation of organic sulfate in the mixed system with the participation of $SO_2$ contributes to the SOA yield. Moreover, as shown in Fig. S9 and Fig. S10, some $C_8$-$C_{10}$ multifunctional groups of di-substituted HOMs could be generated in the mixed systems, thus contributing to the SOA formation. Some compounds showed a trend consistent with the yield at different concentration gradients of DMS, all of which showed that the abundance increased first and then decreased with the increase of DMS concentration, and these results could visually prove that the addition of DMS made the turning point of SOA yield in the mixed systems. However, we were unable to detect products generated by $O_3$ oxidation. We speculate that this is because the reactants do not react with $O_3$ at as high a rate as other oxidizers.

## 4 Conclusions and outlook

Our study contributes to the assessment of the potential impact of marine VOC interaction with SOA formation in ambient air, especially in coastal areas. Under the atmospherically relevant mixing ratio of $\alpha$-pinene to DMS and RH (30-40%), nonlinear effects of DMS on $\alpha$-pinene SOA generation were found. The turning point occurred when the ratio of $\Delta$[DMS] to $\Delta$[$\alpha$-pinene] approached 1.

Combined with the MCM model, we simulated the variation of OH with time during the reaction, which explained well the reason for the appearance of the turning point. The joint action of [OH]$_{ave}$ and sulfur-containing substances adequately explained the rising trend before the turning point. The OH regeneration from the isomerization of $CH_3SCH_2O_2$, an intermediate radical in the oxidation of DMS, could enhance the concentration of OH in the mixed experiments. On the other hand, DMS oxidized to generate highly acidic products that promote or contribute to the mass concentration of particle through rapid nucleation, acid-catalyzed non-homogeneous reactions, and the formation of organosulfates. However, OH reactivity increased after the turning point, which was not conducive to multiple generational VOC oxidation, leading to a slower oxidation process. Low volatile intermediates required more oxidation steps to form particle phase products.

The analysis of SOA chemical composition showed that the effect of DMS on $\alpha$-pinene composition was consistent with the yield variation. $H_2SO_4$ from DMS photooxidation and oxidation products from $\alpha$-pinene interacted to form organosulfates, which contributed to the increase in SOA yield. The addition of DMS enabled the generation of some $C_8$-$C_{10}$ multifunctional groups of di-substituted HOMs in the mixed systems, and these HOMs showed a trend of abundance consistent with the yield as the concentration of DMS increased. Combined with the Van Krevelen plots, we also found that molecular composition of particles in mixed systems had higher and more widely distributed O:C. The high oxidant level in the mixed system at medium DMS concentration helped to enhance the oxidation of the products and increased the O:C ratio of particles. We suggest that the addition of sulfur-containing molecules in the mixed systems reduces the volatility of the products and leads to ELVOC formation, i.e., organosulfates, could promote SOA formation.

This study provides insights into the formation of SOA from the interaction between marine BVOCs and terrestrial BVOCs interaction. We found that the changes in the oxidation state and oxidation pathways of SOA under mixed VOC



conditions need to be further fully investigated. This implies that SOA formation in mixed systems may undergo a highly complex chemical process between sulfur-containing VOC and cyclic VOC degradation chemistry. The parameterization of the interaction effects in the model with respect to VOC mixing then requires additional work and the ability to expand the

modeling of atmospherically relevant conversion from gas-phase species to SOA. In addition to photochemistry, $NO_3$ oxidation at night may also be a major pathway for SOA formation in some cases. However, chemistry of $NO_3$ oxidation of mixed VOCs is not yet known. Therefore, experiments with mixed VOCs need to be carried out comprehensively to further explore the interactions between VOC mixtures and their combined effects on SOA formation. A wider range of experimental conditions should also be considered in the experiments to better represent the real atmospheric conditions.


**Data availability.** Experimental data are available upon request to the corresponding authors.

**Supplement.** The supplement related to this article is available online.

**Author contributions.** LD, KL, and XC designed the experiments and XC carried them out. XC performed data analysis with assistance from KL, LD, ZY, SZ, NTT and JL. XC and KL wrote the paper with contributions from all co-authors.

**Declaration**. The authors declare that they have no conflict of interest.

**Acknowledgements.** We thank Guannan Lin, Jingyao Qu and Zhifeng Li from the State Key Laboratory of Microbial Technology of Shandong University for help and guidance with MS measurements. We thank Wang Xiang from Institute of Chemistry, Chinese Academy of Sciences for the help and guidance with MCM model building.

**Financial support.** This research has been supported by the National Natural Science Foundation of China (grant no.
22376121, 42305100) and Natural Science Foundation of Shandong Province (ZR2023QD092).

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
