# Peer review of "Interaction between marine and terrestrial biogenic volatile organic compounds: Non-linear effect on secondary organic aerosol formation"

_EGUsphere, 2023_

## Author Comment (AC1)

**Response to Anonymous Referee #1**

We are grateful to the Anonymous Referee #1 for the detailed comments and suggestions which greatly improved the quality of our manuscript. Our manuscript has been revised according to the comments from the Referee and our responses to the comments are as follows. Reviewer comments have been copied (R:) and replied to (A:) below. For clarity, the comments are reproduced in blue, authors' responses are in black and changes in the manuscript are in red.

General comments:
This manuscript investigated the SOA formation from a mixture of $\alpha$-pinene and DMS in laboratory chamber experiments. It is found that DMS has a non-linear effect on SOA generation: the mass concentration and yield of SOA show an increasing and then decreasing trend with the increase of the initial concentration of DMS. Potential interaction mechanisms have been proposed. Detailed offline characterization of SOA composition was conducted and utilized to investigate the SOA formation mechanism. However, the analysis has fundamental flaws. I cannot recommend publication in its current form.

Major comments:
R1-1:) The SOA yield is a function of existing organic aerosol (delta_Mo) (Pankow, 1994). This fundamental concept is key in explaining the observed results, but is completely ignored throughout the discussion.

A1-1:) Thank you for your valuable suggestion. As you said, establishing a functional relationship between SOA yield and existing organic aerosol ($\Delta$Mo) is key to interpreting the experimental results. We have added a series of experiments to establish a quantitative relationship between SOA yield and $\Delta$Mo for either $\alpha$-pinene or DMS, as shown in Tables R1 and R2 below. All experiments were conducted at temperature of $299 \pm 1$ K and relative humidity (RH) of $30 - 40\%$. The corresponding contents are added in the revised manuscript (Page 10, Lines 216-217) and supplement (SI Pages 3-4, Lines 48-58; SI Page 20, Lines 291-294).

Table R1. Complementary experiments on the oxidation of individual $\alpha$-pinene.

| Exp. No. | $[\alpha\text{-pinene}]_0$ ppb | $\Delta[\alpha\text{-pinene}]$ ppb | $[NO]_0$ ppb | $[NO_x]_0$ ppb | [SOA] $\mu g\ m^{-3}$ | Y |
|---|---|---|---|---|---|---|
| A'-1 | 111 | 111 | 186 | 197 | 73.8 | 0.12±0.01 |
| A'-2 | 199 | 199 | 196 | 205 | 191.8 | 0.17±0.02 |
| A'-3 | 317 | 317 | 203 | 211 | 269.6 | 0.15±0.02 |
| A'-4 | 415 | 415 | 199 | 207 | 410.7 | 0.18±0.02 |
| A'-5 | 506 | 506 | 174 | 186 | 830.8 | 0.30±0.03 |

Table R2. Complementary experiments on the oxidation of individual DMS.

| Exp. | [DMS]$_0$ | $\Delta$[DMS] | [NO]$_0$ | [NO$_x$]$_0$ | [Total particles] | [H$_2$SO$_4$] | [MSA] | [SOA] | Y |
|------|-----------|---------------|----------|--------------|-------------------|---------------|-------|-------|---|
| No. | ppb | ppb | ppb | ppb | µg m$^{-3}$ | µg m$^{-3}$ | µg m$^{-3}$ | µg m$^{-3}$ | |
| D'-1 | 90 | 90 | 207 | 215 | 50.5 | 11.0 | 8.0 | 22.0 | 0.09±0.01 |
| D'-2 | 161 | 159 | 198 | 207 | 171.1 | 43.5 | 55.2 | 85.9 | 0.21±0.02 |
| D'-3 | 275 | 275 | 199 | 209 | 599.7 | 98.9 | 39.9 | 278.1 | 0.40±0.04 |
| D'-4 | 292 | 265 | 212 | 222 | 556.9 | 97.4 | 150.2 | 295.6 | 0.44±0.04 |

A semi-empirical model based on gas-particle partitioning can be used to describe the relationship between yield and mass loading (Odum et al., 1996; Pankow, 1994), which can be expressed by the following equation (R1):

$$Y = M_O \sum_i \frac{\alpha_i K_{om,i}}{1 + K_{om,i} M_O} \tag{R1}$$

where $\alpha_i$ and $K_{om,i}$ (m$^3$ µg$^{-1}$) are the mass-based gas-phase stoichiometric fraction and gas-particle partitioning coefficient, respectively. Y is the SOA yield, and $\Delta M_O$ is the formed SOA mass concentration. In this model, it is generally assumed that the VOC is converted into a higher volatility product and a lower volatility product. The relationship curve between SOA mass concentration and yield in the experiment can be obtained by fitting the semi-empirical model. The parameters of the base group with lower product volatility were also calculated as $\alpha_1$, $K_{om,1}$, and the parameters of the base group with higher volatility are $\alpha_2$, $K_{om,2}$.

The fitted curves are shown in Fig. R1 below. The DMS fitting curves were also plotted including Exp. D-1. For the individual oxidation experiments of $\alpha$-pinene, the parameters were as follows: $R^2_A = 0.99$, $\alpha_{1,A} = 0.630$, $K_{om,1,A} = 5.178 \times 10^{-4}$, $\alpha_{2,A} = 0.0951$, $K_{om,2,A} = 0.0624$. For DMS, the parameters were as follows: $R^2_D = 0.99$, $\alpha_{1,D} = 1.076$, $K_{om,1,D} = 0.00141$, $\alpha_{2,D} = 0.111$, $K_{om,2,D} = 0.0623$. The $M_O$-dependent SOA yields of $\alpha$-pinene and DMS were also used to calculate the predicted SOA mass concentration in the mixed experiments, **see our response to Comment R1-3.**

[Figure]

Figure R1. Mass-dependent SOA yields for the oxidation of $\alpha$-pinene (a) and DMS (b), individually.

We have also added relevant text in the revised manuscript. As follows:

"A semi-empirical model based on gas-particle partitioning can be used to describe the relationship between yield and mass loading (Odum et al., 1996; Pankow, 1994). The calculation process is shown in Sect. S3 in the supplement. Experiments for both individual $\alpha$-pinene and individual DMS were simulated using this model. The complementary experiments for the two VOCs are shown in Tables S2 and S3. The DMS fitting curves were also plotted including Exp. D-1. The fitted curves are shown in Fig. 3. For the individual oxidation experiments of $\alpha$-pinene, the parameters were as follows: $R^2_A = 0.99$, $\alpha_{1, A} = 0.630$, $K_{om, 1, A} = 5.178 \times 10^{-4}$, $\alpha_{2, A} = 0.0951$, $K_{om, 2, A} = 0.0624$. For DMS, $R^2_D = 0.99$, $\alpha_{1, D} = 1.076$, $K_{om, 1, D} = 0.00141$, $\alpha_{2, D} = 0.111$, $K_{om, 2, D} = 0.0623$." (Pages 9-10, Lines 201-207)

R1-2:) In Fig. 2, there is clear difference between measured and modeled DMS, but this issue is not discussed in the manuscript. The difference is surprising given the $\alpha$-pinene decay is reasonably modeled. Perhaps the DMS measurement has issues. Further, the difference challenges the reliability of modeling results (e.g., Figure 2) and any conclusions drawn based on modeling.

A1-2:) Thank you for your valuable suggestion. First, we think that the DMS measurement is accurate. In our experiments, DMS was detected using GC-FID. We utilized the area of peak at 1.37-1.5 min to quantify DMS, and the calibration curve is shown in Fig. R2 below. There is a good linear correlation between the peak area and the DMS concentration ($R^2 = 0.991$).

[Figure]

Figure R2. Calibration curve of DMS concentration.

Secondly, the large difference between the measured DMS and the modeled DMS is observed in the mixed experiments. This is likely due to the incompleteness of the MCM model for the oxidation mechanism of DMS. We have mentioned the difference in the revised supplement. We elaborate on this reason here as well. The imperfection of the DMS oxidation mechanism in the model and the fact that most studies only focus

on the oxidation mechanism of individual species and lack the mechanism of interaction from the overall perspective result in incomplete agreement of the model simulations (Coates and Butler, 2015; Knote et al., 2015; Zong et al., 2018; Yang et al., 2022). In addition, the MCM DMS scheme suffers from a number of problems. Firstly, unlike the other VOCs simulated by the MCM (alkanes, alkenes, aromatics, and oxygenates), the DMS scheme has rarely evaluated against chamber experiments. We have incorporated the oxidation mechanism of autoxidation of $CH_3SCH_2O_2$ into the MCM model (Table R3) (Berndt et al., 2020; Ye et al., 2022; Jernigan et al., 2022; Lv et al., 2019; Jacob et al., 2024; Chen et al., 2021; Berndt et al., 2023; Veres et al., 2020; Assaf et al., 2023). However, the MCM DMS scheme is rather outdated (Jacob et al., 2024). In addition, the uncertainty in the gas-phase reaction rate constants of the products of DMS and DMS (Chen and Jang, 2012). The corresponding contents are added in the revised supplement (SI Page 5, Lines 93-102; SI Pages 22-23, Line 300).

Table R3. Mechanisms related to DMS added to the MCM model.

| Reaction | Rate constant |
|---|---|
| $CH_3SCH_2O_2 = HOOCH_2SCH_2O_2$ | $2.39 \times 10^9 \times e^{-7278/T}$ |
| $HOOCH_2SCH_2O_2 = HPMTF + OH$ | $6.10 \times 10^{11} \times e^{-9.5 \times 10^3/T + 1.1 \times 10^8/T^3}$ |
| $HOOCH_2SCH_2O_2 + NO = HOOCH_2SCH_2O + NO_2$ | $4.90 \times 10^{-12} \times e^{260/T}$ |
| $HOOCH_2SCH_2O_2 + HO_2 = HOOCH_2SCH_2OOH$ | $KRO2HO2 \times 0.387$ |
| $HOOCH_2SCH_2O_2 + NO_3 = HOOCH_2SCH_2O + NO_2$ | $KRO2NO3$ |
| $HOOCH_2SCH_2O_2 = HOOCH_2SCH_2O$ | $3.74 \times 10^{-12} \times [RO_2] \times 0.8$ |
| $HOOCH_2SCH_2O_2 = HOOCH_2SCH_2OH$ | $3.74 \times 10^{-12} \times [RO_2] \times 0.91$ |
| $HOOCH_2SCH_2O_2 = HPMTF$ | $3.74 \times 10^{-12} \times [RO_2] \times 0.09$ |
| $HOOCH_2SCH_2O = HOOCH_2S + HCHO$ | $1.00 \times 10^6$ |
| $HPMTF + OH = HOOCH_2S + CO$ | $1.75 \times 10^{-11} \times 0.91$ |
| $HPMTF + OH = OH + HCHO + OCS$ | $1.75 \times 10^{-11} \times 0.09$ |
| $HPMTF = HOOCH_2S + HO_2 + CO$ | $2.10 \times 10^{-11}$ |
| $HOOCH_2SCH_2OH + OH = HPMTF + HO_2$ | $2.78 \times 10^{-11}$ |
| $HOOCH_2SCH_2OOH + OH = HOOCH_2SCH_2O_2$ | $2.00 \times 3.68 \times 10^{-13} \times e^{635/T}$ |
| $OCS + O = CO + SO$ | $2.10 \times 10^{-11} \times e^{-2200/T}$ |
| $OCS + OH = SO + OH$ | $7.20 \times 10^{-14} \times e^{-1070/T}$ |
| $SO = SO_2 + O$ | $1.60 \times 10^{-13} \times e^{-2280/T} \times [O_2]$ |
| $SO + O_3 = SO_2$ | $3.40 \times 10^{-12} \times e^{-1100/T}$ |
| $SO + NO_2 = SO_2 + NO$ | $1.40 \times 10^{-11}$ |
| $SO + OH = SO_2 + HO_2$ | $2.60 \times 10^{-11} \times e^{330/T}$ |
| $HOOCH_2S + O_3 = HOOCH_2SO$ | $1.50 \times 10^{-12} \times e^{360/T}$ |
| $HOOCH_2S + NO_2 = HOOCH_2SO + NO$ | $3.00 \times 10^{-11} \times e^{240/T}$ |
| $HOOCH_2S = HOOCH_2SOO$ | $1.20 \times 10^{-16} \times e^{1580/T} \times [O_2]$ |
| $HOOCH_2SOO = TPA + HO_2$ | $7.13 \times 10^{-31} \times T^{14.02} \times e^{-2556/T}$ |
| $HOOCH_2SOO = HOOCH_2S$ | $1.50 \times 10^5$ |
| $HOOCH_2SOO = SO_2 + HCHO + OH$ | $5.00$ |
| $TPA + OH = OCS + OH$ | $5.00 \times 10^{-11} \times 0.14$ |
| $TPA + OH = OCHSOH + OH$ | $5.00 \times 10^{-11} \times 0.86$ |

| Reaction | Rate constant |
|---|---|
| $OCHSOH + OH = OCS + OH$ | $1.40 \times 10^{-12}$ |
| $HOOCH_2SO + O_3 = SO_2 + HCHO + OH$ | $4.00 \times 10^{-13}$ |
| $HOOCH_2SO + NO_2 = SO_2 + HCHO + OH + NO$ | $1.20 \times 10^{-11}$ |
| $OCH_2SCH_2OH = HOCH_2S + HCHO$ | $1.00 \times 10^{6}$ |
| $HOCH_2S + O_3 = HOCH_2SO$ | $1.50 \times 10^{-12} \times e^{360/T}$ |
| $HOCH_2S + NO_2 = HOCH_2SO + NO$ | $3.00 \times 10^{-11} \times e^{240/T}$ |
| $HOCH_2S = HOCH_2SOO$ | $1.20 \times 10^{-16} \times e^{1580/T} \times [O_2]$ |
| $HOCH_2SOO = HOCH_2S$ | $1.50 \times 10^{5}$ |
| $HOCH_2SOO = SO_2 + HCHO + HO_2$ | $5.00$ |
| $HOCH_2SO + O_3 = SO_2 + HCHO + HO_2$ | $4.00 \times 10^{-13}$ |
| $HOCH_2SO + NO_2 = SO_2 + HCHO + HO_2 + NO$ | $1.20 \times 10^{-11}$ |
| $OCH_2SCHO = HCHO + OCS + HO_2$ | $1.00 \times 10^{6}$ |

The following texts were also added in the revised manuscript.

"We have also added the reaction equations and rates obtained from previous studies on the isomerization reaction of the DMS-generated $CH_3SCH_2O_2$ radical (Berndt et al., 2020; Ye et al., 2022; Jernigan et al., 2022; Lv et al., 2019; Jacob et al., 2024; Chen et al., 2021; Berndt et al., 2023; Veres et al., 2020; Assaf et al., 2023) (Table S5)." (Page 7, Lines 167–170)

"In addition, we also fitted the consumption trends of VOCs with the MCM model. There is some deviation between the measured DMS and the fitted DMS in mixed systems. The reasons for the deviation are detailed in Sect. S5 of the supplement. The time series of inorganic gases and the related presentation of the connection with SOA formation are also presented in Sect. S5 of the supplement." (Pages 8-9, Lines 194–197)

Thirdly, the results presented in Fig. 2 of the original manuscript (now Fig. 4c, d in the revised manuscript) are all measured results rather than modeled results. We presented the calculation of average OH concentration and OH reactivity in the supplement (Sect. S6 on Page 6). We have added the relevant texts in the manuscript as well. As follows:

"$[OH]_{avg}$ and OHR were estimated from experimental measurements of VOC concentrations and their OH reaction rate constants. Detailed calculations are given in Sect. S6." (Page 11, Lines 231-233)

Overall, the use of MCM is only supportive. We use this model for the purpose of getting the trend of OH changes in different systems. The vast majority of the results in the manuscript are measured. Although MCM mechanism of DMS is not well established, we believe that the modelled reactivity trend could be used for the comparison with measurements and therefore provide some hints.

R1-3:) The first paragraph under section 2. The effect of $\alpha$-pinene + DMS interaction on SOA yield should be systematically evaluated for all experiments and illustrated graphically. It is not sufficient to compare one set of experiments only in words. Also,

an alternative and more meaningful way is to compare [delta $\alpha$-pinene] * SOA yield$_{\alpha\text{-}pinene}$ + [delta DMS] * SOA yield$_{DMS}$ vs SOA mass formed in the mixed experiments. The SOA yields should correspond to the total SOA mass in the mixed experiments.

A1-3:) Thank you for your valuable suggestion. We calculated the predicted mass concentration for mixed experiments as shown in equations (R2) and (R3) below:

$$\Delta M_i = \Delta VOC_i Y_i \tag{R2}$$

$$\Delta M_o = \sum_{i=1}^{n} \Delta M_i \tag{R3}$$

where $Y_i$ denotes the predicted value calculated using the semi-empirical model when the value of $\Delta M_O$ is equal to that of the mixed experiments, for both $\alpha$-pinene and DMS. $\Delta VOC_i$ ($\mu g\ m^{-3}$) denotes the consumption of $\alpha$-pinene or DMS in the mixed system (Exp. AD-3). $\Delta M_i$ denotes the corresponding predicted mass concentrations of the two VOCs ($\mu g\ m^{-3}$), and $\Delta M_0$ denotes the sum of the predicted mass concentrations of the mixed VOCs ($\mu g\ m^{-3}$).

A comparison of the predicted and measured mass concentration were shown in Fig. R3 below. Consistent with the measured values, the predicted SOA mass concentration showed a turning point at $\Delta$[DMS]/$\Delta$[$\alpha$-pinene] of 1.097. The measured SOA mass concentration was greater than predicted SOA mass concentration for $\Delta$[DMS]/$\Delta$[$\alpha$-pinene] below 1.056. In this ratio range, DMS promoted the generation of SOA in the mixed systems. When the ratio was higher than 1.056, the measured SOA mass concentration was lower than the predicted SOA mass concentration, indicating a possible inhibition effect. The corresponding contents are added in the revised manuscript (Page 10, Lines 210-215) and revised supplement (SI Page 4, Lines 59-63).

[Figure]

Figure R3. Comparison of predicted and measured SOA mass concentration in mixed systems. (a) Measured and predicted SOA mass concentrations in different mixed experiments. (b) Ratio of measured to predicted SOA mass concentration.

The following texts in Table 2 were added in the revised manuscript (Page 9, Line 198).

Table 2. Experimental results of particle-phase components in photooxidation of DMS/$\alpha$-pinene/NO$_x$ systems.

| Exp. No. | [Total particles] [a] $\mu g\ m^{-3}$ | [H$_2$SO$_4$][b] $\mu g\ m^{-3}$ | [MSA][b] $\mu g\ m^{-3}$ | [SOA$_m$][c] $\mu g\ m^{-3}$ | Y$_m$[d] | [SOA$_p$][e] $\mu g\ m^{-3}$ |
|---|---|---|---|---|---|---|
| individual $\alpha$-pinene | | | | | | |
| A-1 | 269.5 | - | - | 269.5 | 0.16±0.02 | |
| individual DMS | | | | | | |
| D-1 | 177.2 | 50.8 | 32.83 | 116.2 | 0.25±0.03 | |
| mix $\alpha$-pinene and DMS | | | | | | |
| AD-1 | 296.3 | 15.0 | 22.2 | 270.8 | 0.14±0.02 | 216.7 |
| AD-2 | 422.3 | 15.7 | 22.5 | 400.1 | 0.20±0.02 | 270.3 |
| AD-3 | 572.6 | 45.4 | 24.9 | 507.5 | 0.24±0.02 | 425.8 |
| AD-4 | 714.4 | 55.4 | 50.5 | 607.7 | 0.25±0.03 | 648.7 |
| AD-5 | 683.0 | 48.8 | 36.2 | 613.1 | 0.24±0.02 | 708.2 |
| AD-6 | 551.5 | 35.3 | 8.5 | 504.2 | 0.19±0.02 | 680.9 |
| AD-7 | 539.9 | 48.4 | 16.8 | 476.0 | 0.19±0.02 | 677.5 |
| AD-8 | 364.4 | 68.7 | 0.1 | 237.1 | 0.08±0.01 | 537.6 |
| AD-9 | 289.9 | 83.2 | 7.0 | 154.0 | 0.06±0.01 | 436.5 |

[a] The mass concentration of particles generated by SMPS, corrected for particle wall loss, was calculated as a particle density of 1.2 g cm$^{-3}$.
[b] IC detection, particle-phase products generated by DMS photooxidation. NH$_4^+$ was hardly detected. All SO$_4^{2-}$ were detected by IC as H$_2$SO$_4$.
[c] The measured SOA mass concentration is expressed as [Total particles]$_{after-correction}$ × (1 - [H$_2$SO$_4$] / [Total particles]$_{before-correction}$).
[d] [SOA$_m$] / ($\Delta$[$\alpha$-pinene] + $\Delta$[DMS]), as mixed yield. Error bars indicate SMPS instrument error of 10%.
[e] The predicted SOA mass concentration by using mass-dependent SOA yields of $\alpha$-pinene and DMS.

We have also added additional texts to the revised manuscript regarding the calculation of predicted mass concentration:

"We evaluated the predicted mass concentration of SOA for two individual systems by using this model. The calculations took place under the condition that the existing organic aerosol ($\Delta$Mo) of the mixed system was equal to that of the individual system. The predicted mass concentration for the different mixed systems were calculated using equations (7) and (8) in the Sect. S3." (Page 10, Lines 207-210)

R1-4:) The proposed explanation regarding the effects of adding DMS on OH concentration is confusing. If the initial OH increase is because of OH regeneration from DMS oxidation, how could it be possible that further adding DMS will reduce OH?

A1-4:) Thank you for your valuable suggestion. The CH$_3$SCH$_2$O$_2$ radical generated from DMS oxidation reacts with NO, RO$_2$, and HO$_2$, and also undergoes isomerization reactions to form OH (Jacob et al., 2024; Berndt et al., 2023; Ye et al., 2022). We have

added reactions related to the isomerization pathway of the $CH_3SCH_2O_2$ radical to the MCM model, **see our answer to Comment R1-2 for details**.

We evaluate the absolute amount of the isomerization channel of the $CH_3SCH_2O_2$ radical using the MCM model. The corresponding contents are added in the revised manuscript (Page 12, Lines 267-272; Page 13, Lines 273-277, 284-286) and supplement (SI Pages 7-8, Lines 128-141). The parameters related to each reaction channel of the $CH_3SCH_2O_2$ radical generated by DMS oxidation were calculated as shown in equations (R4) - (R9):

$$v_{CH_3SCH_2O_2+X,t} = k_{CH_3SCH_2O_2+X}[X]_t \tag{R4}$$

$$C_{CH_3SCH_2O_2+X,t} = \frac{v_{CH_3SCH_2O_2+X,t}}{v_{Isom.} + \sum_{X=NO/RO_2/HO_2} v_{CH_3SCH_2O_2+X,t}} \tag{R5}$$

$$C_{Isom.t} = \frac{v_{Isom.}}{v_{Isom.} + \sum_{X=NO/RO_2/HO_2} v_{CH_3SCH_2O_2+X,t}} \tag{R6}$$

$$\text{Percentage of } CH_3SCH_2O_2+X' \text{ channel} = \frac{\sum_{t=0} C_{CH_3SCH_2O_2+X,t}[CH_3SCH_2O_2]_t}{\sum_{t=0} C_{Isom.t}[CH_3SCH_2O_2]_t + \sum_{X=NO/RO_2/HO_2}(\sum_{t=0} C_{CH_3SCH_2O_2+X,t}[CH_3SCH_2O_2]_t} \tag{R7}$$

$$\text{Percentage of Isom.' channel} = \frac{\sum_{t=0} C_{Isom.t}[CH_3SCH_2O_2]_t}{\sum_{t=0} C_{Isom.t}[CH_3SCH_2O_2]_t + \sum_{X=NO/RO_2/HO_2}(\sum_{t=0} C_{CH_3SCH_2O_2+X,t}[CH_3SCH_2O_2]_t} \tag{R8}$$

$$\text{Amount of Isom.} = \sum_{t=0} C_{CH_3SCH_2O_2+X,t}[CH_3SCH_2O_2]_t \tag{R9}$$

where X denotes the concentration of NO, RO$_2$, or HO$_2$ (molecule cm$^{-3}$) at time t fitted by the MCM model in each experiment. $v_{CH3SCH2O2+X,t}$ denotes the rate (s$^{-1}$) at which the bimolecular reaction ($CH_3SCH_2O_2$ + $NO/RO_2/HO_2$) at time t, respectively. $k_{CH3SCH2O2+X}$ denotes the rate constant (molecule cm$^{-3}$ s$^{-1}$) for the reaction of the $CH_3SCH_2O_2$ radical with NO, RO$_2$ or HO$_2$ at time t, respectively. The rate constants are respectively (Jacob et al., 2024): $k_{CH3SCH2O2+NO}$ = $1.169 \times 10^{-10}$ molecule cm$^{-3}$ s$^{-1}$, $k_{CH3SCH2O2+RO2}$ = $3.740 \times 10^{-12}$ molecule cm$^{-3}$ s$^{-1}$, $k_{CH3SCH2O2+HO2}$ = $5.805 \times 10^{-12}$ molecule cm$^{-3}$ s$^{-1}$. $v_{Isom.}$ is a constant, here assumed to be 0.06 s$^{-1}$ (Jacob et al., 2024; Assaf et al., 2023). $C_{CH3SCH2O2+X,t}$ (%) denotes the rate percentage of the three bimolecular reaction channels at time t. $C_{Isom.t}$ (%) denotes the rate percentage of the isomerization reaction channel. $[CH_3SCH_2O_2]_t$ (molecule cm$^{-3}$) denotes the concentration of $CH_3SCH_2O_2$ radical at moment t. The percentage of $CH_3SCH_2O_2$ + X'channel or Isom.'channel (%) indicates the relative percentage of a particular bimolecular or isomerization reaction channel throughout the whole reaction process. Amount of Isom. (molecule cm$^{-3}$) denotes the absolute amount of the isomerization channel throughout the reaction process.

The simulation results are shown in Fig. R4 below. It can be found that the amount of the isomerization channel increases and then decreases as the ratio of precursor consumption increases. The increase in OHR mentioned in the manuscript leads to a decreasing trend of SOA yield after the turning point. This is directly supported by the amount of the isomerization channel of the model-fitted $CH_3SCH_2O_2$ radical. As the

Δ[DMS]/Δ[α-pinene] increases further, the absolute amount of isomerization decreases and OH regeneration is less significant.

This estimation result agrees with the measured SOA mass concentration, SOA yield, and OH concentration trends showing in Fig. 4, with the turning point at the Δ[DMS]/Δ[α-pinene] ratio of ~0.6 - 1 (i.e., Exp. AD-3 or AD-4). The slight difference (i.e., turning point at AD-3 vs AD-4) is likely due to the incomplete mechanism for DMS in the MCM model. Nevertheless, the results here suggest that the isomerization reaction intensity controls the OH concentration and therefore SOA formation in the mixed experiments.

[Figure]

Figure R4. Amount of isomerization channels of $CH_3SCH_2O_2$ radical. A curve was drawn as a guide to the eye. The curve was fitted without using the last data point since it was much higher than the other points.

R1-5:) The proposed mechanisms in Figures 8 and 10 are flawed. The proposed isomerization reactions and H-shift do not occur in the atmosphere (Xu et al., 2019; Vereecken and Nozière, 2020).

A1-5:) Thank you for your valuable suggestion. We agree that the H-shift path shown in Fig. 8 from the original manuscript indeed cannot occur after reviewing the literature. We have removed the second pathway in Fig. 8 from the original manuscript after revision. At the same time, we have modified the first pathway to other pathways, while proposing $C_{10}H_{15}NO_6$ to be the other isomer. The molecular structure of the modified $C_{10}H_{15}NO_6$ is a ring-opening product, which is oxidized from pinonaldehyde (Eddingsaas et al., 2012a). In addition, we retained the molecular structure of the dimer $C_{20}H_{33}NO_8$, which is derived from Draper et al. (2015). The following texts and figures were added in the revised manuscript.

"Figure 12b shows the possible pathway of ON formation. In the presence of $NO_2$, the hydrogen atoms on the carbon chain of the typical product pinonaldehyde can be readily oxidized to form nitrogen-containing carboxylate products by the addition of oxygen, i.e., $C_{10}H_{15}NO_6$ (MW 245) (Boyd et al., 2015; Kim et al., 2012; Eddingsaas et al., 2012b)." (Page 20, Lines 417-420)

"In addition, Fig. 12b demonstrates the possible structure of a high molecular weight oligomer generated in the individual $\alpha$-pinene experiments: $C_{20}H_{33}NO_8$ (MW 415). It is speculated that $RO_2$ tends more towards isomerization processes such as autoxidation compared to fragmentation reaction (Draper et al., 2015). This pathway increases the possibility of oligomerization of $RO_2 + RO_2$ and $RO_2 + HO_2$ in individual $\alpha$-pinene oxidation." (Page 20, Lines 423-426)

The mechanisms related to ONs have also been modified, as shown below (Page 21, Lines 427-430).

[Figure]

Figure 12. Proposed formation mechanisms and structural for organosulfate (a) and organic nitrates (b) in SOA. Red, blue and black in the boxes refer to the products identified by $\alpha$-pinene-only SOA products, mixed-only SOA products and $\alpha$-pinene-mixed-both SOA products, respectively.

We have combined Fig. 9, 10, and 11 from the original manuscript and modified the relevant textual content. The most direct effect of DMS on $\alpha$-pinene SOA is the generation of sulfur-containing molecules. The fragmentation product $C_5H_8O_8S$ can be generated from pinonic acid. This has been demonstrated in Surratt et al. (2008). For $C_{10}H_{17}NO_7S$, we have modified it based on Wang et al. (2020). This product forms $RO_2$ by addition of OH and $O_2$. Then ON is formed in the presence of NO, followed by esterification by sulfuric acid to form the final product. As for $C_{10}H_{16}N_2O_{10}S$ and $C_6H_{10}O_8S$, we also found their presence in the studies of Surratt et al. (2008) and Xu et al. (2020). The relevant mechanisms of OS are shown in Fig. 12(a) above. The following texts were added in the revised manuscript.

"Acidic products of DMS can participate in the formation of $\alpha$-pinene SOA. Acidic products are mainly generated by the H-abstraction pathway via photooxidation of DMS. At room temperature, H abstraction is the main reaction pathway, accounting for ~70% of the total reaction (Barnes et al., 2006). As shown in Fig. 1, this pathway first generates the $CH_3SCH_2$ radical, which can add to $O_2$ to generate the $CH_3SCH_2O_2$ radical, followed by reaction with $RO_2$ or $HO_2$ to generate peroxy hydroxyl or peroxy nitrate, or with NO, eventually generating $SO_2$, MSA and $H_2SO_4$. In the composition analysis of SOA generated by photooxidation of DMS-$\alpha$-pinene, a number of sulfur-containing compounds were generated. Based on previous studies (Aschmann et al., 2002; Surratt et al., 2008; Surratt et al., 2007; Gao et al., 2006), we identify some potential OSs (Table S4). Figure 12a shows the reaction pathway for $C_5$ OS generation. The oxidation product, pinonic acid reacts with OH to form 2/3-hydroxyglutaric acid (3-Hydroxyglutaric acid as an example) under high-NO condition, followed by sulfation to form $C_5H_8O_8S$ (MW 228) (Surratt et al., 2008). Xu et al. (2020) detected the isomer of $C_6H_{10}O_8S$ (MW 242), which was speculated to be a fragmentation product formed by further oxidation by ring structure breakage during the reaction." (Page 20, Lines 396-406)

"Due to the presence of $NO_x$, we detected monomer compounds containing nitrogen-containing OSs (NOSs) which can demonstrate the coexistence of nitrate and sulfate groups. Figure 12a shows the reaction pathway for one of $C_{10}$ NOS generation. $C_{10}H_{17}NO_7S$ is formed by heterogeneous reactions of $SO_2$ with intermediates. The pathway usually occurs in the particle phase (Wang et al., 2020). Surratt et al. (2008) had identified the $C_{10}H_{16}N_2O_{10}S$ (MW 356) product and proposed that it would form in the presence of highly acidic seeds that produces hydroperoxides containing two nitro groups firstly, which further eliminate one water molecule by sulfation. Combined with Table S4, the changes in the relative abundance of the majority of OSs in different mixing systems are consistent with the pattern of change in the yield of mixed SOA. This phenomenon suggests that DMS is most likely to influence the mechanism of $\alpha$-pinene SOA production through acidic products, especially the contribution of the $C_{10}$ NOSs." (Page 20, Lines 407-415)

We have found that DMS also affects the CHO molecules of $\alpha$-pinene SOA. We have speculated on the mechanisms of formation of some CHO molecules. This part was presented in Sect. S8, Fig. S9 and Fig. S10 of the original supplement. We selected

representative substances therein, mainly highly oxygenated organic molecules of $C_8$ - $C_{10}$. They are $C_8H_{12}O_4$, $C_7H_{10}O_4$, $C_{10}H_{16}O_2$, $C_{10}H_{16}O_3$, $C_{10}H_{16}O_4$, $C_9H_{14}O_4$, $C_9H_{14}O_6$, $C_8H_{12}O_3$, $C_9H_{14}O_5$, $C_8H_{12}O_6$, and $C_8H_{12}O_5$, respectively. These products represent $\alpha$-pinene-only SOA products, mixed-only SOA products and $\alpha$-pinene-mixed-both SOA products, respectively. Among them, the three products, $C_{10}H_{16}O_4$, $C_9H_{14}O_6$, and $C_8H_{12}O_5$, have similar trends in the variation of yield in different systems.

The following texts and figure were added in the revised manuscript.

"The effect of DMS on the formation of $\alpha$-pinene SOA is multiple. Table S4 shows some typical substances in different oxidation systems. The addition of DMS resulted in the generation of more $C_8$ - $C_{10}$ HOMs in the $\alpha$-pinene SOA. Based on the previous studies (Librando and Tringali, 2005; Kristensen et al., 2014; Yasmeen et al., 2010; Aschmann et al., 1998; Gao et al., 2006), we showed the possible formation pathways of some of these typical CHO molecules in Fig. 11. In the presence of $NO_x$, $\alpha$-pinene undergoes OH addition and oxidation to form $RO_2$. In the first pathway, this first-generational $RO_2$ undergoes multigenerational autoxidation or reacts with NO to form terpenylic acid ($C_8H_{12}O_4$, MW 172). Terpenylic acid has the highest relative abundance in the product of $\alpha$-pinene SOA compared to the mixed systems. Therefore, in an individual system, terpenylic acid is more likely to undergo further oxidation to produce terebic acid ($C_7H_{10}O_4$, MW 158), which proves the view of Claeys et al. (2009) that the onset of terebic aldehyde oxidation is assumed to be in the particle phase, and this substance contributes dominantly to the particle phase product in individual $\alpha$-pinene oxidation." (Pages 17-18, Lines 373-382)

"In the second pathway, the first-generational $RO_2$ can interact with NO to form RO, which can undergo isomerization and other processes to form pinonaldehyde ($C_{10}H_{16}O_2$, MW 168). Pinonaldehyde can be accompanied by the generation of $HO_2$, and then further oxidized to hydroperoxide ($C_{10}H_{16}O_4$, MW 200) and $cis$-pinonic acid ($C_{10}H_{16}O_3$, MW 184), which is a typical gas-phase product. $cis$-Pinonic acid can produce $RO_2$ containing several functional groups, which forms pinic acid ($C_9H_{14}O_4$, MW 186) in the presence of NO and $HO_2$. Further oxidation of pinic acid can form $RO_2$, which undergoes reactions such as isomerization, dissociation to form a number of monomer compounds of $C_8$ and $C_9$ in the particle phase, including the HOMs such as $C_9H_{14}O_6$ (MW 218), $C_9H_{14}O_5$ (MW 202), $C_8H_{12}O_3$ (MW 156). $C_8H_{12}O_3$ oxidized by chain termination to from $C_8H_{12}O_5$ (MW 188) and $C_8H_{12}O_6$ (MW 204). Overall, the addition of DMS promotes a deeper oxidation of the intermediate product generated by the oxidation of $\alpha$-pinene, as reflected in the opening of the six-membered ring, resulting in the formation of a multifunctional peroxycarboxylic acid." (Pages 18-19, Lines 383-392)

[Figure]

Figure 11. Proposed formation mechanisms and structural for CHO molecules in SOA. Red, blue and black in the boxes refer to the products identified by α-pinene-only SOA products, mixed-only SOA products and α-pinene-mixed-both SOA products, respectively.

Subsequently, we also proposed the mechanisms for the other CHO products in the text of the revised supplement.

"Several other CHO compounds were also generated in different systems. Based on the previous studies (Librando and Tringali, 2005; Kristensen et al., 2014; Yasmeen

et al., 2010; Aschmann et al., 1998; Gao et al., 2006), we sorted out the possible formation mechanisms of the products of individual $\alpha$-pinene and mixed oxidation in our experiments, as shown in Fig. S9. Figure S9 illustrates the four photooxidation pathways of $\alpha$-pinene. $\alpha$-Pinene forms $RO_2$ under the oxidation of OH and $O_2$, and further forms RO in the presence of $RO_2$ or NO. The six-membered ring of RO opens, removing a $CH_3$ fragment molecule to form the product, namely $C_9H_{14}O_3$ (MW 170). Further oxidation of the gas-phase typical product, pinic acid, leads to the formation of $C_8H_{12}O_3$ (MW 156). This product undergoes the action of $O_2$ to form $RO_2$, which passes through the $RO_2$ + $HO_2$ channel to form the final product, $C_8H_{12}O_4$ (MW 172). In addition to generating hydroperoxide ($C_{10}H_{16}O_4$, MW 200) and $cis$-pinonic acid ($C_{10}H_{16}O_3$, MW 184) as mentioned in the Section 3.3.2, the intermediate product pinonaldehyde can be oxidized to form $C_9H_{14}O_2$ (MW 154) and $C_{10}H_{18}O_3$ (MW 186)." (SI Page 10, Lines 205-214)

"Figure S9 also shows other products formed by the oxidation of the gas-phase product $cis$-pinonic acid ($C_{10}H_{16}O_3$, MW 184) besides the products in Fig. 11, which forms a number of peroxyl radicals in the presence of OH or $O_2$, and the subsequent reactions of these peroxyl radicals can be divided into four pathways." (SI Page 10, Lines 215-217)

Revised possible reaction pathway of CHO products is showed as Fig. S9 in the supplement (SI Page 17, Lines 270-272).

[Figure]

Figure S9. Possible reaction pathway of CHO products generation. Blue and black in the boxes refer to the product identified by mixed-only SOA products and $\alpha$-pinene-mixed-both SOA products, respectively.

Minor Comments:
R1-6:) The head row of Table 3 is confusing. The table seems to have two different head rows. For example, does the first column correspond to [total particles] or delta[$\alpha$-pinene]?

A1-6:) Thank you for your valuable suggestion. We have modified the head row of Table 3 (now Table 2 in the revised manuscript).
Page 9, Line 198 in the revised manuscript. As follows:

Table 2. Experimental results of particle-phase components in photooxidation of DMS/$\alpha$-pinene/NO$_x$ systems.

| Exp. No. | [Total particles] [a] μg m$^{-3}$ | [H$_2$SO$_4$] [b] μg m$^{-3}$ | [MSA] [b] μg m$^{-3}$ | [SOA$_m$] [c] μg m$^{-3}$ | Y$_m$ [d] | [SOA$_p$] [e] μg m$^{-3}$ |
|---|---|---|---|---|---|---|
| **individual $\alpha$-pinene** | | | | | | |
| A-1 | 269.5 | - | - | 269.5 | 0.16±0.02 | |
| **individual DMS** | | | | | | |
| D-1 | 177.2 | 50.8 | 32.83 | 116.2 | 0.25±0.03 | |
| **mix $\alpha$-pinene and DMS** | | | | | | |
| AD-1 | 296.3 | 15.0 | 22.2 | 270.8 | 0.14±0.02 | 216.7 |
| AD-2 | 422.3 | 15.7 | 22.5 | 400.1 | 0.20±0.02 | 270.3 |
| AD-3 | 572.6 | 45.4 | 24.9 | 507.5 | 0.24±0.02 | 425.8 |
| AD-4 | 714.4 | 55.4 | 50.5 | 607.7 | 0.25±0.03 | 648.7 |
| AD-5 | 683.0 | 48.8 | 36.2 | 613.1 | 0.24±0.02 | 708.2 |
| AD-6 | 551.5 | 35.3 | 8.5 | 504.2 | 0.19±0.02 | 680.9 |
| AD-7 | 539.9 | 48.4 | 16.8 | 476.0 | 0.19±0.02 | 677.5 |
| AD-8 | 364.4 | 68.7 | 0.1 | 237.1 | 0.08±0.01 | 537.6 |
| AD-9 | 289.9 | 83.2 | 7.0 | 154.0 | 0.06±0.01 | 436.5 |

[a] The mass concentration of particles generated by SMPS, corrected for particle wall loss, was calculated as a particle density of 1.2 g cm$^{-3}$.
[b] IC detection, particle-phase products generated by DMS photooxidation. NH$_4^+$ was hardly detected. All SO$_4^{2-}$ were detected by IC as H$_2$SO$_4$.
[c] The measured SOA mass concentration is expressed as [Total particles]$_{after-correction}$ × (1 - [H$_2$SO$_4$] / [Total particles]$_{before-correction}$).
[d] [SOA$_m$] / ($\Delta$[$\alpha$-pinene] + $\Delta$[DMS]), as mixed yield. Error bars indicate SMPS instrument error of 10%.
[e] The predicted SOA mass concentration by using mass-dependent SOA yields of $\alpha$-pinene and DMS.

R1-7:) Describe how the volatility of each compound is estimated for Figure 6.

A1-7:) Thank you for your valuable suggestion. We showed how to assess the volatility of each compound in the supplement. The following texts were added in the revised manuscript.
    "The volatility calculation for all the components are shown in Sect. S4." (Page 15, Lines 332-333)

R1-8:) Line 3. Grammar error. I assume what the authors want to express is that "OH generation before the turning point could attribute to the enhancement in SOA formation."

A1-8:) Thank you for your valuable suggestion. The following texts were revised in the new manuscript.

"OH regeneration before the turning point could attribute to the enhancement in SOA formation." (Page 12, Lines 250-251)

References:

Aschmann, S. M., Atkinson, R., and Arey, J.: Products of reaction of OH radicals with $\alpha$-pinene, J. Geophys. Res., 107, ACH 6-1-ACH 6-7, 10.1029/2001JD001098, 2002.

Aschmann, S. M., Reisseil, A., Atkinson, R., and Arey, J.: Products of the gas phase reactions of the OH radical with $\alpha$- and $\beta$-pinene in the presence of NO, J. Geophys. Res., 103, 25553-25561, 10.1029/98JD01676, 1998.

Assaf, E., Finewax, Z., Marshall, P., Veres, P. R., Neuman, J. A., and Burkholder, J. B.: Measurement of the intramolecular hydrogen-shift rate coefficient for the $CH_3SCH_2OO$ radical between 314 and 433 K, J. Phys. Chem. A, 127, 2336-2350, 10.1021/acs.jpca.2c09095, 2023.

Barnes, I., Hjorth, J., and Mihalopoulos, N.: Dimethyl sulfide and dimethyl sulfoxide and their oxidation in the atmosphere, Chem. Rev., 106, 940-975, 10.1021/cr020529+, 2006.

Berndt, T., Hoffmann, E. H., Tilgner, A., Stratmann, F., and Herrmann, H.: Direct sulfuric acid formation from the gas-phase oxidation of reduced-sulfur compounds, Nat. Commun., 14, 4849, 10.1038/s41467-023-40586-2, 2023.

Berndt, T., Chen, J., Møller, K. H., Hyttinen, N., Prisle, N. L., Tilgner, A., Hoffmann, E. H., Herrmann, H., and Kjaergaard, H. G.: $SO_2$ formation and peroxy radical isomerization in the atmospheric reaction of OH radicals with dimethyl disulfide, Chem. Comm., 56, 13634-13637, 10.1039/D0CC05783E, 2020.

Boyd, C. M., Sanchez, J., Xu, L., Eugene, A. J., Nah, T., Tuet, W. Y., Guzman, M. I., and Ng, N. L.: Secondary organic aerosol formation from the $\beta$-pinene + $NO_3$ system: Effect of humidity and peroxy radical fate, Atmos. Chem. Phys., 15, 7497-7522, 10.5194/acp-15-7497-2015, 2015.

Chen, J., Berndt, T., Møller, K. H., Lane, J. R., and Kjaergaard, H. G.: Atmospheric fate of the $CH_3SOO$ radical from the $CH_3S$ + $O_2$ equilibrium, J. Phys. Chem. A, 125, 8933-8941, 10.1021/acs.jpca.1c06900, 2021.

Claeys, M., Iinuma, Y., Szmigielski, R., Surratt, J. D., Blockhuys, F., Van Alsenoy, C., Böge, O., Sierau, B., Gómez-González, Y., Vermeylen, R., Van der Veken, P., Shahgholi, M., Chan, A. W. H., Herrmann, H., Seinfeld, J. H., and Maenhaut, W.: Terpenylic acid and related compounds from the oxidation of $\alpha$-pinene: Implications for new particle formation and growth above forests, Environ. Sci. Technol., 43, 6976-6982, 10.1021/es9007596, 2009.

Coates, J. and Butler, T. M.: A comparison of chemical mechanisms using tagged ozone production potential (TOPP) analysis, Atmos. Chem. Phys., 15, 8795-8808, 10.5194/acp-15-8795-2015, 2015.

Draper, D. C., Farmer, D. K., Desyaterik, Y., and Fry, J. L.: A qualitative comparison of secondary organic aerosol yields and composition from ozonolysis of monoterpenes at varying concentrations of $NO_2$, Atmos. Chem. Phys., 15, 12267-12281, 10.5194/acp-15-12267-2015, 2015.

Eddingsaas, N. C., Loza, C. L., Yee, L. D., Seinfeld, J. H., and Wennberg, P. O.: $\alpha$-Pinene photooxidation under controlled chemical conditions – Part 1: Gas-phase composition in low- and high-$NO_x$ environments, Atmos. Chem. Phys., 12, 6489-6504, 10.5194/acp-12-6489-2012, 2012a.

Eddingsaas, N. C., Loza, C. L., Yee, L. D., Chan, M., Schilling, K. A., Chhabra, P. S., Seinfeld, J. H., and Wennberg, P. O.: $\alpha$-Pinene photooxidation under controlled chemical conditions – Part 2: SOA yield and composition in low- and high-$NO_x$ environments, Atmos. Chem. Phys., 12, 7413-7427, 10.5194/acp-12-7413-2012, 2012b.

Gao, S., Surratt, J. D., Knipping, E. M., Edgerton, E. S., Shahgholi, M., and Seinfeld, J. H.: Characterization of polar organic components in fine aerosols in the southeastern United States: Identity, origin, and evolution, J. Geophys. Res., 111, D14314, 10.1029/2005JD006601, 2006.

Jacob, L. S. D., Giorio, C., and Archibald, A. T.: Extension, development, and evaluation of the representation of the OH-initiated dimethyl sulfide (DMS) oxidation mechanism in the Master Chemical Mechanism (MCM) v3.3.1 framework, Atmos. Chem. Phys., 24, 3329-3347, 10.5194/acp-24-3329-2024, 2024.

Jernigan, C. M., Fite, C. H., Vereecken, L., Berkelhammer, M. B., Rollins, A. W., Rickly, P. S., Novelli, A., Taraborrelli, D., Holmes, C. D., and Bertram, T. H.: Efficient production of carbonyl sulfide in the low-$NO_x$ oxidation of dimethyl sulfide, Geophys. Res. Lett., 49, e2021GL096838, 10.1029/2021GL096838, 2022.

Kim, H., Barkey, B., and Paulson, S. E.: Real refractive indices and formation yields of secondary organic aerosol generated from photooxidation of limonene and $\alpha$-pinene: The effect of the HC/$NO_x$ ratio, J. Phys. Chem. A, 116, 6059–6067, 10.1021/jp301302z.s001, 2012.

Knote, C., Tuccella, P., Curci, G., Emmons, L., Orlando, J. J., Madronich, S., Baró, R., Jiménez-Guerrero, P., Luecken, D., Hogrefe, C., Forkel, R., Werhahn, J., Hirtl, M., Pérez, J. L., San José, R., Giordano, L., Brunner, D., Yahya, K., and Zhang, Y.: Influence of the choice of gas-phase mechanism on predictions of key gaseous pollutants during the AQMEII phase-2 intercomparison, Atmos. Environ., 115, 553-568, 10.1016/j.atmosenv.2014.11.066, 2015.

Kristensen, K., Cui, T., Zhang, H., Gold, A., Glasius, M., and Surratt, J. D.: Dimers in $\alpha$-pinene secondary organic aerosol: Effect of hydroxyl radical, ozone, relative humidity and aerosol acidity, Atmos. Chem. Phys., 14, 4201-4218, 10.5194/acp-14-4201-2014, 2014.

Librando, V. and Tringali, G.: Atmospheric fate of OH initiated oxidation of terpenes. Reaction mechanism of $\alpha$-pinene degradation and secondary organic aerosol formation, J. Environ. Manage., 75, 275-282, 10.1016/j.jenvman.2005.01.001, 2005.

Lv, G., Zhang, C., and Sun, X.: Understanding the oxidation mechanism of methanesulfinic acid by ozone in the atmosphere, Sci. Rep., 9, 322, 10.1038/s41598-018-36405-0, 2019.

Odum, J. R., Hoffmann, T., Bowman, F., Collins, D., Flagan, R. C., and Seinfeld, J. H.: Gas/Particle partitioning and Secondary Organic Aerosol Yields, Environ. Sci. Technol., 30, 2580-2585, 10.1021/es950943+, 1996.

Pankow, J. F.: An absorption model of gas/particle partitioning of organic compounds in the atmosphere, Atmos. Environ., 28, 185-188, 10.1016/1352-2310(94)90093-0, 1994.

Surratt, J. D., Kroll, J. H., Kleindienst, T. E., Edney, E. O., Claeys, M., Sorooshian, A., Ng, N. L., Offenberg, J. H., Lewandowski, M., Jaoui, M., Flagan, R. C., and Seinfeld, J. H.: Evidence for organosulfates in secondary organic aerosol, Environ. Sci. Technol., 41, 517-527, 10.1021/es062081q, 2007.

Surratt, J. D., Gómez-González, Y., Chan, A. W. H., Vermeylen, R., Shahgholi, M., Kleindienst, T. E., Edney, E. O., Offenberg, J. H., Lewandowski, M., Jaoui, M., Maenhaut, W., Claeys, M., Flagan, R. C., and Seinfeld, J. H.: Organosulfate formation in biogenic secondary organic aerosol, J. Phys. Chem. A, 112, 8345-8378, 10.1021/jp802310p, 2008.

Vereecken, L. and Nozière, B.: H migration in peroxy radicals under atmospheric conditions, Atmos. Chem. Phys., 20, 7429-7458, 10.5194/acp-20-7429-2020, 2020.

Veres, P. R., Neuman, J. A., Bertram, T. H., Assaf, E., Wolfe, G. M., Williamson, C. J., Weinzierl, B., Tilmes, S., Thompson, C. R., Thames, A. B., Schroder, J. C., Saiz-Lopez, A., Rollins, A. W., Roberts, J. M., Price, D., Peischl, J., Nault, B. A., Møller, K. H., Miller, D. O., Meinardi, S., Li, Q., Lamarque, J.-F., Kupc, A., Kjaergaard, H. G., Kinnison, D., Jimenez, J. L., Jernigan, C. M., Hornbrook, R. S., Hills, A., Dollner, M., Day, D. A., Cuevas, C. A., Campuzano-Jost, P., Burkholder, J., Bui, T. P., Brune, W. H., Brown, S. S., Brock, C. A., Bourgeois, I., Blake, D. R., Apel, E. C., and Ryerson, T. B.: Global airborne sampling reveals a previously unobserved dimethyl sulfide oxidation mechanism in the marine atmosphere, Proc. Natl. Acad. Sci., 117, 4505-4510, 10.1073/pnas.1919344117, 2020.

Wang, Y., Hu, M., Wang, Y. C., Li, X., Fang, X., Tang, R., Lu, S., Wu, Y., Guo, S., Wu, Z., Hallquist, M., and Yu, J. Z.: Comparative study of particulate organosulfates in contrasting atmospheric environments: Field evidence for the significant influence of anthropogenic sulfate and $NO_x$, Environ. Sci. Technol. Lett., 7, 787-794, 10.1021/acs.estlett.0c00550, 2020.

Xu, L., Møller, K. H., Crounse, J. D., Otkjær, R. V., Kjaergaard, H. G., and Wennberg, P. O.: Unimolecular reactions of peroxy radicals formed in the oxidation of $\alpha$-pinene and $\beta$-pinene by hydroxyl radicals, J. Phys. Chem. A, 123, 1661-1674, 10.1021/acs.jpca.8b11726, 2019.

Xu, L., Tsona, N. T., You, B., Zhang, Y., Wang, S., Yang, Z., Xue, L., and Du, L.: $NO_x$ enhances secondary organic aerosol formation from nighttime $\gamma$-terpinene ozonolysis, Atmos. Environ., 225, 117375, 10.1016/j.atmosenv.2020.117375, 2020.

Yang, X., Yuan, B., Peng, Z., Peng, Y., Wu, C., Yang, S., Li, J., and Shao, M.: Inter-comparisons of VOC oxidation mechanisms based on box model: A focus on OH reactivity, J. Environ. Sci., 114, 286-296, 10.1016/j.jes.2021.09.002, 2022.

Yasmeen, F., Vermeylen, R., Szmigielski, R., Iinuma, Y., Böge, O., Herrmann, H., Maenhaut, W., and Claeys, M.: Terpenylic acid and related compounds: precursors for dimers in secondary organic aerosol from the ozonolysis of $\alpha$- and $\beta$- pinene, Atmos. Chem. Phys., 10, 9383-9392, 10.5194/acp-10-9383-2010, 2010.

Ye, Q., Goss, M. B., Krechmer, J. E., Majluf, F., Zaytsev, A., Li, Y., Roscioli, J. R., Canagaratna, M., Keutsch, F. N., Heald, C. L., and Kroll, J. H.: Product distribution, kinetics, and aerosol formation from the OH oxidation of dimethyl sulfide under different $RO_2$ regimes, Atmos. Chem. Phys., 22, 16003-16015, 10.5194/acp-22-16003-2022, 2022.

Zong, R., Xue, L., Wang, T., and Wang, W.: Inter-comparison of the regional atmospheric chemistry mechanism (RACM2) and master chemical mechanism (MCM) on the simulation of acetaldehyde, Atmos. Environ., 186, 144-149, 10.1016/j.atmosenv.2018.05.013, 2018.

---

## Author Comment (AC2)

**Response to Anonymous Referee #2**

We are grateful to the Anonymous Referee #2 for the detailed comments and suggestions which greatly improved the quality of our manuscript. Our manuscript has been revised according to the comments from the Referee and our responses to the comments are as follows. Reviewer comments have been copied (R:) and replied to (A:) below. For clarity, the comments are reproduced in blue, authors' responses are in black and changes in the manuscript are in red.

General comments:
This manuscript describes a system of environmental chamber experiments to examine the kinetics and product distribution both in the gas and aerosol phase from a mixture of DMS and $\alpha$-pinene. The work describes the non-linear effect of DMS on the oxidation of $\alpha$-pinene with respect to the mass concentration and yield of SOA. The authors attribute this observation primarily to acid catalyzed heterogeneous reactions and changing OH reactivity and concentrations. The authors present a detailed analysis of the SOA and the components that could be contributing to the observed SOA. The authors also present multiple mechanisms to help explain the observed masses. The work in its current form is confusing and contains errors and issues with the figures and supporting claims. I cannot recommend this publication in its current form. I would request major revisions before publication is reevaluated.

Major comments
R2-1:) The tables and figures throughout the work are confusing. Axes are hard to associate with the data and tables seem to have headers that do not match with the presented data.

A2-1:) Thank you for your valuable suggestion.
The following texts, tables and figures were revised in the new manuscript.
Page 4, Lines 101-102:
    All experiments were conducted at temperature of 299 ± 1 K and relative humidity (RH) of 30 – 40%.

Page 5, Line 111:
Table 1. Experimental conditions of the chamber experiments.

| Exp. No. | $[\alpha\text{-pinene}]_0^a$ ppb | $[DMS]_0^a$ ppb | $\Delta[\alpha\text{-pinene}]^b$ ppb | $\Delta[DMS]^b$ ppb | $[NO]_0^a$ ppb | $[NO_x]_0^a$ ppb | $[SO_2]_{max}^c$ ppb | $[O_3]_{max}^c$ ppb |
|---|---|---|---|---|---|---|---|---|
| | | | | **individual $\alpha$-pinene** | | | | |
| A-1* | 308 | 0 | 308 | - | 195 | 206 | - | 21 |
| A-2* | 285 | 0 | 285 | - | 201 | 204 | - | 30 |
| | | | | **individual DMS** | | | | |
| D-1 | 0 | 184 | - | 183 | 189 | 192 | 68 | 19 |
| D-2* | 0 | 290 | - | 276 | 206 | 211 | 103 | 25 |
| D-3* | 0 | 600 | - | 372 | 207 | 212 | 183 | 41 |

| | | | | mix α-pinene and DMS | | | | |
|---|---|---|---|---|---|---|---|---|
| AD-1 | 312 | 140 | 312 | 83 | 208 | 211 | 35 | 25 |
| AD-2 | 321 | 197 | 321 | 87 | 190 | 195 | 42 | 25 |
| AD-3 | 305 | 301 | 305 | 181 | 203 | 212 | 76 | 23 |
| AD-4 | 291 | 372 | 291 | 307 | 183 | 203 | 89 | 18 |
| AD-5 | 308 | 441 | 308 | 338 | 193 | 202 | 111 | 25 |
| AD-6 | 317 | 536 | 317 | 359 | 191 | 203 | 110 | 24 |
| AD-7 | 282 | 639 | 282 | 384 | 196 | 206 | 140 | 27 |
| AD-8 | 306 | 613 | 306 | 440 | 189 | 193 | 156 | 45 |
| AD-9 | 295 | 687 | 295 | 457 | 183 | 191 | 154 | 51 |
| AD-10* | 319 | 251 | 319 | 219 | 197 | 200 | 82 | 25 |
| AD-11* | 314 | 401 | 314 | 330 | 184 | 194 | 116 | 22 |
| AD-12* | 332 | 646 | 332 | 447 | 198 | 199 | 179 | 26 |
| AD-13* | 300 | 614 | 300 | 406 | 193 | 197 | 147 | 29 |

[a] Initial concentration of α-pinene, DMS, NO and $NO_x$.
[b] The consumption of α-pinene and DMS when the particles were produced to the maximum mass concentration determined by SMPS.
[c] The maximum concentration of $O_3$ and $SO_2$ production during light exposure.
* For off-line analysis of SOA. MS Analysis: Exp. A-1, D-2, AD-10, AD-11, AD-12. IR Analysis: Exp. A-2, D-3, AD-13.

Page 9, Line 198:

Table 2. Experimental results of particle-phase components in photooxidation of DMS/α-pinene/$NO_x$ systems.

| Exp. No. | [Total particles] [a] μg m$^{-3}$ | [$H_2SO_4$] [b] μg m$^{-3}$ | [MSA] [b] μg m$^{-3}$ | [$SOA_m$] [c] μg m$^{-3}$ | $Y_m$ [d] | [$SOA_p$] [e] μg m$^{-3}$ |
|---|---|---|---|---|---|---|
| | | | individual α-pinene | | | |
| A-1 | 269.5 | - | - | 269.5 | 0.16±0.02 | |
| | | | individual DMS | | | |
| D-1 | 177.2 | 50.8 | 32.83 | 116.2 | 0.25±0.03 | |
| | | | mix α-pinene and DMS | | | |
| AD-1 | 296.3 | 15.0 | 22.2 | 270.8 | 0.14±0.02 | 216.7 |
| AD-2 | 422.3 | 15.7 | 22.5 | 400.1 | 0.20±0.02 | 270.3 |
| AD-3 | 572.6 | 45.4 | 24.9 | 507.5 | 0.24±0.02 | 425.8 |
| AD-4 | 714.4 | 55.4 | 50.5 | 607.7 | 0.25±0.03 | 648.7 |
| AD-5 | 683.0 | 48.8 | 36.2 | 613.1 | 0.24±0.02 | 708.2 |
| AD-6 | 551.5 | 35.3 | 8.5 | 504.2 | 0.19±0.02 | 680.9 |
| AD-7 | 539.9 | 48.4 | 16.8 | 476.0 | 0.19±0.02 | 677.5 |
| AD-8 | 364.4 | 68.7 | 0.1 | 237.1 | 0.08±0.01 | 537.6 |
| AD-9 | 289.9 | 83.2 | 7.0 | 154.0 | 0.06±0.01 | 436.5 |

[a] The mass concentration of particles generated by SMPS, corrected for particle wall loss, was calculated as a particle density of 1.2 g cm$^{-3}$.
[b] IC detection, particle-phase products generated by DMS photooxidation. $NH_4^+$ was hardly detected. All $SO_4^{2-}$ were detected by IC as $H_2SO_4$.
[c] The measured SOA mass concentration is expressed as [Total particles]$_{after-correction}$ × (1 - [$H_2SO_4$] / [Total particles]$_{before-correction}$).
[d] [$SOA_m$] / (Δ[α-pinene] + Δ[DMS]), as mixed yield. Error bars indicate SMPS instrument error of 10%.

Page 7, Line 173:

Table 2 show the experimental results of particle-phase components in photooxidation of DMS and α-pinene systems.

Page 8, Lines 185-187:

[Figure]

Figure 2. Variation of precursors with reaction time. Red and black dots indicate the results of smog chamber experiments and the curves indicate the results of MCM simulations. Blue dots indicate mass concentration of particles in smog chamber.

Page 14, Lines 299-303:

[Figure]

Figure 6. Comparison of negative mode mass peak of SOA in the individual α-pinene and mixed systems. (a) Mass spectra of SOA with/without the presence of DMS. (b) Comparison of the relative intensities of mass spectrometry peaks with different *m/z*

ratio ranges. Relative strength is the strength of a substance with a certain mass-to-charge ratio divided by the total strength of all substances.

Page 15, Lines 329-331:

[Figure]

Figure 8. The plots of OS$_C$ against carbon number of particulate organic molecules formed from individual α-pinene (a) and mixed conditions (b).

Page 16, Lines 341-343:

[Figure]

Figure 9. Relationship between O:C and saturation concentration for molecules of different element types in individual α-pinene oxidation (a) and mixed oxidation (b).

Page 17, Lines 360-365:

[Figure]

Figure 10. Relative abundance of molecules identified in different reaction systems. (a) S-containing components in the mixed oxidation at different DMS concentrations. (b) CHO/CHON components recognized in individual and mixed oxidation. Details of the molecules are given in Table S4. "Low, medium and high" represent $\Delta[DMS]/\Delta[\alpha$-pinene] in mixed experiments, indicating different mixing ratios. Specifically, "low" represents Exp. AD-10 below the turning point, "medium" represents Exp. AD-11 at the turning point, and "high" represents Exp. AD-12 below the turning point.

The following figures were added or revised in the new supplement.
SI Page 13, Lines 250-253:

[Figure]

Figure S2. Wall-loss correction of particle. (a) Variation of particle wall-loss coefficient with particle size. (b) Total particle mass concentration of aerosols generated from mixed system. (c) Size-dependent particle number concentration before correction from mixed system. (d) Size-dependent particle number concentration after correction from mixed system.

SI Page 17, Lines 268-269:

[Figure]

Figure S8. Van Krevelen plots of compounds formed from individual α-pinene (a) and mixed system (b).

R2-2:) The work does a good job trying to disentangle the effect of DMS and alpha-pinene on SOA formation by looking at the products and yields separately as well as in a mixture. The concentrations of VOCs, $NO_x$ and $H_2O_2$ are atypical of the environment and should be discussed in more detail. I understand limitations of instrumentation and observations, but additional work (facilitated by modeling) could be used to better understand the fate of the VOC and the $RO_2$ generated from OH oxidation within the chamber.

A2-2:) Thank you for your valuable suggestion. We agree that the fate of the $RO_2$ generated from OH oxidation is important to be understood in the chamber. Therefore, we evaluate the fate of $RO_2$ by estimating the relative contribution (percentage) of $RO_2$ + $HO_2$, $RO_2$ + NO and $RO_2$ + $RO_2$ reactions. In the quantitative calculations for these three channels, we used the recommended general rate constants (Ziemann and Atkinson, 2012). Although the type of $RO_2$ and the product channels can bias the results slightly, it is negligible for overall quantification (Peng et al., 2019). We focus on $RO_2$ in general and do not specifically discuss the chemical characterization of specific $RO_2$ oxidation products. It is important to note in particular that the $RO_2$ self- and cross-reaction rate constant is highly correlated with $RO_2$ type (Peng et al., 2019). Rate constants are highly dependent on the specific $RO_2$ types and can vary over a very large range ($10^{-17}$ - $10^{-10}$ $cm^3$ molecule$^{-1}$s$^{-1}$). Unsubstituted primary, secondary and tertiary $RO_2$ radicals self- and cross-reaction rate at ~ $10^{-13}$, ~ $10^{-15}$ and ~ $10^{-17}$ $cm^3$ molecule$^{-1}$s$^{-1}$, respectively (Ziemann and Atkinson, 2012). Substituted $RO_2$ types have higher reaction rate constants than unsubstituted $RO_2$ types, which can reach ~$10^{-11}$ $cm^3$ molecule$^{-1}$s$^{-1}$. Based on this, our study used relatively moderate levels of rate constants to quantify $RO_2$ + $RO_2$ channels. A value of $2.5 \times 10^{-13}$ $cm^3$ molecule$^{-1}$s$^{-1}$ was chosen based on Ziemann and Atkinson (2012)'s study. Meanwhile, for the $RO_2$ + $HO_2$ and $RO_2$ + NO channels, we selected the values $1.5 \times 10^{-11}$ $cm^3$ molecule$^{-1}$s$^{-1}$ and $9 \times 10^{-12}$

cm$^3$ molecule$^{-1}$s$^{-1}$ (Ziemann and Atkinson, 2012; Peng et al., 2019). The corresponding contents are added in the revised supplement (SI Pages 8-9, Lines 149-160).

With the known reaction rate constants for each channel, we calculated the contribution of the three reaction channels using the following equations (R1) - (R4):

$$C_{RO_2+HO_2,t} = \frac{k_{RO_2+HO_2}[HO_2]_t}{k_{RO_2+HO_2}[HO_2]_t + k_{RO_2+RO_2}[RO_2]_t + k_{RO_2+NO}[NO]_t} \tag{R1}$$

$$C_{RO_2+RO_2,t} = \frac{k_{RO_2+RO_2}[RO_2]_t}{k_{RO_2+HO_2}[HO_2]_t + k_{RO_2+RO_2}[RO_2]_t + k_{RO_2+NO}[NO]_t} \tag{R2}$$

$$C_{RO_2+NO,t} = \frac{k_{RO_2+NO}[NO]_t}{k_{RO_2+HO_2}[HO_2]_t + k_{RO_2+RO_2}[RO_2]_t + k_{RO_2+NO}[NO]_t} \tag{R3}$$

$$\text{Percentage of each channel} = \frac{\sum_{t=0}[RO_2]_t C_{RO_2+HO_2,t} \text{ or} \sum_{t=0}[RO_2]_t C_{RO_2+RO_2,t} \text{ or} \sum_{t=0}[RO_2]_t C_{RO_2+NO,t}}{\sum_{t=0}[RO_2]_t C_{RO_2+HO_2,t} + \sum_{t=0}[RO_2]_t C_{RO_2+RO_2,t} + \sum_{t=0}[RO_2]_t C_{RO_2+NO,t}} \tag{R4}$$

where $C_{RO2+HO2,t}$, $C_{RO2+RO2,t}$ and $C_{RO2+NO,t}$ (s$^{-1}$) are the rate percentage of the three channels at the given time point (t), respectively. $[HO_2]_t$, $[RO_2]_t$ and $[NO]_t$ (molecule cm$^{-3}$) represent the concentration simulated in the MCM model at t, respectively. The percentage of each channel refers to the relative percentage of each reaction channel throughout the whole reaction process. $k_{RO2+RO2} = 2.5 \times 10^{-13}$ cm$^3$ molecule$^{-1}$ s$^{-1}$, $k_{RO2+HO2} = 1.5 \times 10^{-11}$ cm$^3$ molecule$^{-1}$ s$^{-1}$, $k_{RO2+NO} = 9 \times 10^{-12}$ cm$^3$ molecule$^{-1}$ s$^{-1}$. The corresponding contents are added in the revised supplement (SI Page 8, Lines 142-148)

The calculation results are displayed in Fig. R1a below. The reaction pathway with the largest contribution is RO$_2$ + NO (~50-80%), indicating that our experiments are under the typical high-NO$_x$ conditions. Even though the concentration of VOCs are atypical in the real atmosphere, it can be seen that the percentage of the RO$_2$ + RO$_2$ reaction pathway is very low (<10%). This indicates that the RO$_2$ self- or cross-reaction pathway is a minor fate in our chamber experiments. The corresponding contents are added in the revised supplement (SI Page 9, Lines 161-164; SI Page 18, Lines 273-275).

In addition to the three bimolecular reaction pathways (i.e. RO$_2$ + RO$_2$, RO$_2$ + HO$_2$ and RO$_2$ + NO), RO$_2$ isomerization is also an important reaction pathway. To determine whether isomerization can occur in different reaction systems, we calculated RO$_2$ bimolecular lifetimes ($\tau$) (Xu et al., 2019), as shown in Equation (R5):

$$\tau = \frac{1}{k_{RO_2+HO_2}[HO_2]_t + k_{RO_2+RO_2}[RO_2]_t + k_{RO_2+NO}[NO]_t} \tag{R5}$$

An RO$_2$ lifetime (without RO$_2$ isomerization included) of 10 s leads to a relative importance of isomerization of 50% in the total fate (including all loss pathways) of RO$_2$ with an isomerization rate constant of 0.1s$^{-1}$, which is a typical order of magnitude for isomerization rate constants of multifunctional RO$_2$ with hydroxyl and hydroperoxy substituents (Crounse et al., 2013; D'ambro et al., 2017; Praske et al., 2018). Peng et al. (2019) used models to simulate the atmospheric lifetime of RO$_2$ in several typical ambient sites, chambers and flow tubes, with data extracted from (Fry et al., 2013; Martin et al., 2016; Martin et al., 2017; Ryerson et al., 2013; Peng et al., 2016; Mao et al., 2009; Stone et al., 2012; Nguyen et al., 2014). NO measured in Los Angeles during the CalNex-LA campaign (Ortega et al., 2016) was 1 ppb, which would to allow RO$_2$

to isomerize, even in an urban area. The environmental conditions in this region are similar to our experiments, both in a high-$NO_x$ environment. While 10 s is an important threshold, the conditions that apply are remote clean areas where little $RO_2$ + NO reaction occurs. The atmospheric lifetime of $RO_2$ in the Los Angeles area starts at 0.3 s, which coincides with our experimental. Therefore, we used 0.3 s as the threshold to evaluate whether isomerization can occur under high $NO_x$ conditions. It can be found that the $RO_2$ lifetime of all experimental systems are higher than 0.3 s from the Fig. R1b. Thus, it is likely that the isomerization channel of $RO_2$ can occur in our experiments. The corresponding contents are added in the revised supplement (SI Page 9, Lines 165-180; SI Page 18, Lines 273-275).

Overall, even though the concentration of VOCs and oxidants in our experimental systems are not typical of the environment, the $RO_2$ fate can still be considered atmospherically relevant.

[Figure]

Figure R1. MCM model fitting results for the $RO_2$ reaction channel. (a) Percentage of different reaction channels of $RO_2$ in different experiments. (b) Atmospheric lifetime of $RO_2$ (without $RO_2$ isomerization included) in different oxidation systems.

Meanwhile, the following texts were added in the revised manuscript.

"In addition, even though the concentrations of VOCs and oxidants in our experimental systems are not typical of the environment, our experiments can still be considered atmospherically relevant. As discussed in detail in Sect. S7, the calculation results indicate that our experiments were conducted under typical high $NO_x$ conditions. Although VOC concentrations are high, $RO_2$ + $RO_2$ reaction is not a major fate of $RO_2$. $RO_2$ isomerization can likely occur in our experiments as well." (Page 4, Lines 103-107)

R2-3:) Overall the experimental set up and design of the chamber experiments needs to be discussed in more detail, so the reader can understand the observations better. In particular the design of the chamber is not communicated well. There is some general confusion about the [OH] concentration within the chamber and how that plays into the observations of DMS and $\alpha$-pinene. The DMS observations across the chamber results

seem to have a linear and unmatched decay compared to that of the model. This is a significant fraction of mass that the model is not capturing that is not addressed in the text. A further discussion on limitations or missing mechanisms within the DMS mechanism should be communicated to better understand the present results and subsequent understanding on DMS's effect on SOA yields.

A2-3:) Thank you for your valuable suggestion. We calculated the average OH concentration by the decay of measured DMS concentration in the chamber. We found that the trend of OH concentration in different oxidation systems was consistent with the yield, and all of them showed an increasing and then decreasing trend. The change in OH concentration is likely due to the OH regeneration from the isomerization of $CH_3SCH_2O_2$ radical. The $CH_3SCH_2O_2$ radical generated from DMS oxidation reacts with NO, $RO_2$, and $HO_2$, and also undergoes isomerization reactions to form OH (Jacob et al., 2024; Berndt et al., 2023; Ye et al., 2022). We have added reactions related to the isomerization pathway of the $CH_3SCH_2O_2$ radical to the MCM model (Table R1) (Berndt et al., 2020; Ye et al., 2022; Jernigan et al., 2022; Lv et al., 2019; Jacob et al., 2024; Chen et al., 2021; Berndt et al., 2023; Veres et al., 2020; Assaf et al., 2023).

Table R1. Mechanisms related to DMS added to the MCM model.

| Reaction | Rate constant |
| --- | --- |
| $CH_3SCH_2O_2 = HOOCH_2SCH_2O_2$ | $2.39\times10^9\times e^{-7278/T}$ |
| $HOOCH_2SCH_2O_2 = HPMTF + OH$ | $6.10\times10^{11}\times e^{-9.5\times10^3/T+1.1\times10^8/T^3}$ |
| $HOOCH_2SCH_2O_2 + NO = HOOCH_2SCH_2O + NO_2$ | $4.90\times10^{-12}\times e^{260/T}$ |
| $HOOCH_2SCH_2O_2 + HO_2 = HOOCH_2SCH_2OOH$ | $KRO2HO2\times0.387$ |
| $HOOCH_2SCH_2O_2 + NO_3 = HOOCH_2SCH_2O + NO_2$ | $KRO2NO3$ |
| $HOOCH_2SCH_2O_2 = HOOCH_2SCH_2O$ | $3.74\times10^{-12}\times[RO_2]\times0.8$ |
| $HOOCH_2SCH_2O_2 = HOOCH_2SCH_2OH$ | $3.74\times10^{-12}\times[RO_2]\times0.91$ |
| $HOOCH_2SCH_2O_2 = HPMTF$ | $3.74\times10^{-12}\times[RO_2]\times0.09$ |
| $HOOCH_2SCH_2O = HOOCH_2S + HCHO$ | $1.00\times10^6$ |
| $HPMTF + OH = HOOCH_2S + CO$ | $1.75\times10^{-11}\times0.09$ |
| $HPMTF + OH = OH + HCHO + OCS$ | $1.75\times10^{-11}\times0.92$ |
| $HPMTF = HOOCH_2S + HO_2 + CO$ | $2.10\times10^{-11}$ |
| $HOOCH_2SCH_2OH + OH = HPMTF + HO_2$ | $2.78\times10^{-11}$ |
| $HOOCH_2SCH_2OOH + OH = HOOCH_2SCH_2O_2$ | $2.00\times3.68\times10^{-13}\times e^{635/T}$ |
| $OCS + O = CO + SO$ | $2.10\times10^{-11}\times e^{-2200/T}$ |
| $OCS + OH = SO + OH$ | $7.20\times10^{-14}\times e^{-1070/T}$ |
| $SO = SO_2 + O$ | $1.60\times10^{-13}\times e^{-2280/T}\times[O_2]$ |
| $SO + O_3 = SO_2$ | $3.40\times10^{-12}\times e^{-1100/T}$ |
| $SO + NO_2 = SO_2 + NO$ | $1.40\times10^{-11}$ |
| $SO + OH = SO_2 + HO_2$ | $2.60\times10^{-11}\times e^{330/T}$ |
| $HOOCH_2S + O_3 = HOOCH_2SO$ | $1.50\times10^{-12}\times e^{360/T}$ |
| $HOOCH_2S + NO_2 = HOOCH_2SO + NO$ | $3.00\times10^{-11}\times e^{240/T}$ |
| $HOOCH_2S = HOOCH_2SOO$ | $1.20\times10^{-16}\times e^{1580/T}\times[O_2]$ |
| $HOOCH_2SOO = TPA + HO_2$ | $7.13\times10^{-31}\times T^{14.02}\times e^{-2556/T}$ |

| Reaction | Rate constant |
|---|---|
| $HOOCH_2SOO = HOOCH_2S$ | $1.50 \times 10^5$ |
| $HOOCH_2SOO = SO_2 + HCHO + OH$ | $5.00$ |
| $TPA + OH = OCS + OH$ | $5.00 \times 10^{-11} \times 0.14$ |
| $TPA + OH = OCHSOH + OH$ | $5.00 \times 10^{-11} \times 0.86$ |
| $OCHSOH + OH = OCS + OH$ | $1.40 \times 10^{-12}$ |
| $HOOCH_2SO + O_3 = SO_2 + HCHO + OH$ | $4.00 \times 10^{-13}$ |
| $HOOCH_2SO + NO_2 = SO_2 + HCHO + OH + NO$ | $1.20 \times 10^{-11}$ |
| $OCH_2SCH_2OH = HOCH_2S + HCHO$ | $1.00 \times 10^6$ |
| $HOCH_2S + O_3 = HOCH_2SO$ | $1.50 \times 10^{-12} \times e^{360/T}$ |
| $HOCH_2S + NO_2 = HOCH_2SO + NO$ | $3.00 \times 10^{-11} \times e^{240/T}$ |
| $HOCH_2S = HOCH_2SOO$ | $1.20 \times 10^{-16} \times e^{1580/T} \times [O_2]$ |
| $HOCH_2SOO = HOCH_2S$ | $1.50 \times 10^5$ |
| $HOCH_2SOO = SO_2 + HCHO + HO_2$ | $5.00$ |
| $HOCH_2SO + O_3 = SO_2 + HCHO + HO_2$ | $4.00 \times 10^{-13}$ |
| $HOCH_2SO + NO_2 = SO_2 + HCHO + HO_2 + NO$ | $1.20 \times 10^{-11}$ |
| $OCH_2SCHO = HCHO + OCS + HO_2$ | $1.00 \times 10^6$ |

We evaluate the absolute amount of the isomerization channel of the $CH_3SCH_2O_2$ radical using the MCM model. The corresponding contents are added in the revised manuscript (Page 12, Lines 267-272; Page 13, Lines 273-277, 284-286) and supplement (SI Pages 7-8, Lines 128-141). The parameters related to each reaction channel of the $CH_3SCH_2O_2$ radical generated by DMS oxidation were calculated as shown in equations (R6) - (R11):

$$v_{CH_3SCH_2O_2+X,t} = k_{CH_3SCH_2O_2+X}[X]_t \tag{R6}$$

$$C_{CH_3SCH_2O_2+X,t} = \frac{v_{CH_3SCH_2O_2+X,t}}{v_{Isom.} + \sum\limits_{X=NO/RO_2/HO_2} v_{CH_3SCH_2O_2+X,t}} \tag{R7}$$

$$C_{Isom.t} = \frac{v_{Isom.}}{v_{Isom.} + \sum\limits_{X=NO/RO_2/HO_2} v_{CH_3SCH_2O_2+X,t}} \tag{R8}$$

$$\text{Percentage of } CH_3SCH_2O_2+X' \text{ channel} = \frac{\sum\limits_{t=0} C_{CH_3SCH_2O_2+X,t}[CH_3SCH_2O_2]_t}{\sum\limits_{t=0} C_{Isom.t}[CH_3SCH_2O_2]_t + \sum\limits_{X=NO/RO_2/HO_2} (\sum\limits_{t=0} C_{CH_3SCH_2O_2+X,t}[CH_3SCH_2O_2]_t} \tag{R9}$$

$$\text{Percentage of Isom.' channel} = \frac{\sum\limits_{t=0} C_{Isom.t}[CH_3SCH_2O_2]_t}{\sum\limits_{t=0} C_{Isom.t}[CH_3SCH_2O_2]_t + \sum\limits_{X=NO/RO_2/HO_2} (\sum\limits_{t=0} C_{CH_3SCH_2O_2+X,t}[CH_3SCH_2O_2]_t} \tag{R10}$$

$$\text{Amount of Isom.} = \sum\limits_{t=0} C_{CH_3SCH_2O_2+X,t}[CH_3SCH_2O_2]_t \tag{R11}$$

where X denotes the concentration of NO, $RO_2$, or $HO_2$ (molecule cm$^{-3}$) at time t fitted by the MCM model in each experiment. $v_{CH3SCH2O2+X,t}$ denotes the rate (s$^{-1}$) at which the bimolecular reaction ($CH_3SCH_2O_2$ + NO / $RO_2$ / $HO_2$) at time t, respectively. $k_{CH3SCH2O2+X}$ denotes the rate constant (molecule cm$^{-3}$ s$^{-1}$) for the reaction of the $CH_3SCH_2O_2$ radical with NO, $RO_2$ or $HO_2$ at time t, respectively. The rate constants

are respectively (Jacob et al., 2024): $k_{CH3SCH2O2+NO} = 1.169 \times 10^{-10}$ molecule cm$^{-3}$ s$^{-1}$, $k_{CH3SCH2O2+RO2} = 3.740 \times 10^{-12}$ molecule cm$^{-3}$ s$^{-1}$, $k_{CH3SCH2O2+HO2} = 5.805 \times 10^{-12}$ molecule cm$^{-3}$ s$^{-1}$. $v_{Isom.}$ is a constant, here assumed to be 0.06 s$^{-1}$ (Jacob et al., 2024; Assaf et al., 2023). $C_{CH3SCH2O2+X,t}$ (%) denotes the rate percentage of the three bimolecular reaction channels at time t. $C_{Isom.t}$ (%) denotes the rate percentage of the isomerization reaction channel. $[CH_3SCH_2O_2]_t$ (molecule cm$^{-3}$) denotes the concentration of $CH_3SCH_2O_2$ radical at moment t. The percentage of $CH_3SCH_2O_2$ + X'channel or Isom.'channel (%) indicates the relative percentage of a particular bimolecular or isomerization reaction channel throughout the whole reaction process. Amount of Isom. (molecule cm$^{-3}$) denotes the absolute amount of the isomerization channel throughout the reaction process.

The simulation results are shown in Fig. R2 below. It can be found that the amount of the isomerization channel increases and then decreases as the ratio of precursor consumption increases. The increase in OHR mentioned in the manuscript leads to a decreasing trend of SOA yield after the turning point. This is directly supported by the amount of the isomerization channel of the model-fitted $CH_3SCH_2O_2$ radical. As the $\Delta[DMS]/\Delta[\alpha\text{-pinene}]$ increases further, the absolute amount of isomerization decreases and OH regeneration is less significant.

This estimation result agrees with the measured SOA mass concentration, SOA yield, and OH concentration trends showing in Fig. 4, with the turning point at the $\Delta[DMS]/\Delta[\alpha\text{-pinene}]$ ratio of ~ 0.6 - 1 (i.e., Exp. AD-3 or AD-4). The slight difference (i.e., turning point at AD-3 vs AD-4) is likely due to the incomplete mechanism for DMS in the MCM model. Nevertheless, the results here suggest that the isomerization reaction intensity controls the OH concentration and therefore SOA formation in the mixed experiments.

[Figure]

Figure R2. Amount of isomerization channels of $CH_3SCH_2O_2$ radical. A curve was drawn as a guide to the eye. The curve was fitted without using the last data point since it was much higher than the other points.

Elevated OH concentration leads to faster precursors consumption and more SOA generation. We have added details related to the influence of OH concentration on VOCs and SOA in the revised manuscript. As follows:

"OH regeneration before the turning point could attribute to the enhancement in SOA formation. Increasing the average OH concentration within the reaction system helps to enhance the oxidation rates of $\alpha$-pinene and DMS (Ng et al., 2007), resulting in the rapid generation of low volatile products. The rapid formation of low volatile products ensures the formation and growth of SOA even if there is wall loss of the gas phase products. In addition to this, high OH concentration contributes to multigenerational oxidation reactions (Sarrafzadeh et al., 2016; Robinson et al., 2007; Eddingsaas et al., 2012b). For example, intermediates such as unsaturated keto-aldehydes as well as epoxides can be generated during the photooxidation of $\alpha$-pinene. Increasing the OH concentration of the reaction system consumes more intermediates, resulting in an increase in the SOA mass concentration. The SOA yield was calculated based on the amount of precursors consumed as well as the SOA mass concentration. Thus the obtained SOA yield is higher at high OH concentration." (Page 12, Lines 250-258)

The large difference between the measured DMS and the modeled DMS is observed in the mixed experiments. This is likely due to the incompleteness of the MCM model for the oxidation mechanism of DMS. We have mentioned the difference in the revised supplement. We elaborate on this reason here as well. The imperfection of the DMS oxidation mechanism in the model and the fact that most studies only focus on the oxidation mechanism of individual species and lack the mechanism of interaction from the overall perspective result in incomplete agreement of the model simulations (Coates and Butler, 2015; Knote et al., 2015; Zong et al., 2018; Yang et al., 2022). In addition, the MCM DMS scheme suffers from a number of problems. Unlike the other VOCs simulated by the MCM (alkanes, alkenes, aromatics, and oxygenates), the DMS scheme has rarely evaluated against chamber experiments. We have incorporated the oxidation mechanism of autoxidation of $CH_3SCH_2O_2$ into the MCM model (Table R1) (Berndt et al., 2020; Ye et al., 2022; Jernigan et al., 2022; Lv et al., 2019; Jacob et al., 2024; Chen et al., 2021; Berndt et al., 2023; Veres et al., 2020; Assaf et al., 2023). However, the MCM DMS scheme is rather outdated (Jacob et al., 2024). In addition, the uncertainty in the gas-phase reaction rate constants of the products of DMS and DMS (Chen and Jang, 2012). The corresponding contents are added in the revised supplement (SI Page 5, Lines 93-102; SI Pages 22-23, Line 300).

Overall, the use of MCM is only supportive. We use this model for the purpose of getting the trend of OH changes in different systems. The vast majority of the results in the manuscript are measured. Although MCM mechanism of DMS is not well established, we believe that the modelled reactivity trend could be used for the comparison with measurements and therefore provide some hints.

We have also added a short description of the gap between the fitted and measured DMS in the revised manuscript. As follows:

"In addition, we also fitted the consumption trends of VOCs with the MCM model. There is some deviation between the measured DMS and the fitted DMS in mixed systems. The reasons for the deviation are detailed in Sect. S5 of the supplement. The time series of inorganic gases and the related presentation of the connection with SOA formation are also presented in Sect. S5 of the supplement." (Pages 8-9, Lines 194–197)

In addition, we have revised the calculation of the average OH concentration in the revised manuscript. As follows:
"The steps for calculating the SOA yield have been mentioned earlier in Table 2." (Page 10, Line 227)
"[OH]$_{avg}$ and OHR were estimated from experimental measurements of VOC concentrations, and their OH reaction rate constants. Detailed calculations are given in Sect. S6." ( Page 11, Lines 231-233)

R2-4:) The discussion surrounding OH is confusing and deserves more explanation. In particular, more time needs to be spent describing (or citing) how OH was calculated/constrained within the chamber. Figure 2, presents OH numbers and a trend line, but descriptions of how this was derived is missing. DMS has a well known OH loss rate and is present in most of the experiments, I would recommend using that decay curve to inform your OH concentrations and compare that to your box model results. Additionally, I am confused about the connection between OH loss and aerosol acidity referenced in the work (line 190).

A2-4:) Thank you for your valuable suggestion. **The response regarding OH calculation and the comparison to model can be found above in our response to Comment R2-3.**
We apologize for the confusion caused by the use of the phrase "In addition to acid catalysis". We would like to clarify that DMS affects SOA generation not only through acid catalysis, but also through its own actions such as OH regeneration. In the new manuscript, we have removed this sentence. The trends in OH simulated by the MCM model were described in the original manuscript. We have also presented these texts in the revised manuscript. It is shown below:
"To more accurately reflect the [OH]$_{avg}$ of each experiment, we combined the MCM model to calculate the trends of OH concentration in different experiments with time, as shown in Fig. S6. The maximum OH concentration before the turning point from Exp. AD-1 to Exp. AD-4 are higher than those after the turning point from Exp. AD-5 to Exp. AD-9 at the end of experiments. Interestingly, the largest OH concentration formed in Exp. AD-4 during the time period when the OH concentration was rising at the fastest rate from the local magnification graph, which is consistent with the average OH concentration reflected in Fig. 4d. The OH concentrations of other systems are also largely consistent." (Page 12, Lines 244-250)

R2-5:) Overall, there seems to be a lack of citations or validation for some of the comments made throughout the work. In particular, comprehensive citations

referencing mechanisms, techniques and analysis used and previous chamber work is missing.

A2-5:) Thank you for your valuable suggestion. We add more references to the relevant parts. In addition, the introduction mentioned some work related to chamber for simulated oxidation of mixed VOCs.

Page 2, Lines 36, 37, 38, 42 in the revised manuscript. As follows:

[revised manuscript text omitted]

Voliotis, A., Wang, Y., Shao, Y., Du, M., Bannan, T. J., Percival, C. J., Pandis, S. N., Alfarra, M. R., and McFiggans, G.: Exploring the composition and volatility of secondary organic aerosols in mixed anthropogenic and biogenic precursor systems, Atmos. Chem. Phys., 21, 14251-14273, 10.5194/acp-21-14251-2021, 2021.

In addition, the following references were added in the revised manuscript.

"The chemical composition of SOA was determined by ultra performance liquid chromatography (UPLC, UltiMate 3000, Thermo Scientific) coupled with quadrupole time-of-flight mass spectrometry (Q-TOFMS, Bruker Impact HD) (Zhang et al., 2016)." (Page 6, Line 136)

"OH regeneration before the turning point could attribute to the enhancement in SOA formation. Increasing the average OH concentration within the reaction system

helps to enhance the oxidation rates of $\alpha$-pinene and DMS (Ng et al., 2007), resulting in the rapid generation of low volatile products." (Page 12, Line 252)

"In addition to this, high OH concentration contributes to multigenerational oxidation reactions (Eddingsaas et al., 2012b; Sarrafzadeh et al., 2016; Robinson et al., 2007)." (Page 12, Line 255)

"Based on the previous studies (Librando and Tringali, 2005; Kristensen et al., 2014; Yasmeen et al., 2010; Aschmann et al., 1998; Gao et al., 2006), we show the possible formation pathways of some of these typical CHO molecules in Fig. 11." (Pages 17-18, Lines 374-376)

The cited references are shown below:

Zhang, X., Dalleska, N. F., Huang, D. D., Bates, K. H., Sorooshian, A., Flagan, R. C., and Seinfeld, J. H.: Time-resolved molecular characterization of organic aerosols by PILS + UPLC/ESI-Q-TOFMS, Atmos. Environ., 130, 180-189, 10.1016/j.atmosenv.2015.08.049, 2016.

Ng, N. L., Kroll, J. H., Chan, A. W. H., Chhabra, P. S., Flagan, R. C., and Seinfeld, J. H.: Secondary organic aerosol formation from m-xylene, toluene, and benzene, Atmos. Chem. Phys., 7, 3909-3922, 10.5194/acp-7-3909-2007, 2007.

Eddingsaas, N. C., Loza, C. L., Yee, L. D., Chan, M., Schilling, K. A., Chhabra, P. S., Seinfeld, J. H., and Wennberg, P. O.: $\alpha$-Pinene photooxidation under controlled chemical conditions - Part 2: SOA yield and composition in low- and high-$NO_x$ environments, Atmos. Chem. Phys., 12, 7413-7427, 10.5194/acp-12-7413-2012, 2012b.

Sarrafzadeh, M., Wildt, J., Pullinen, I., Springer, M., Kleist, E., Tillmann, R., Schmitt, S. H., Wu, C., Mentel, T. F., Zhao, D., Hastie, D. R., and Kiendler-Scharr, A.: Impact of $NO_x$ and OH on secondary organic aerosol formation from $\beta$-pinene photooxidation, Atmos. Chem. Phys., 16, 11237-11248, 10.5194/acp-16-11237-2016, 2016.

Robinson, A. L., Donahue, N. M., Shrivastava, M. K., Weitkamp, E. A., Sage, A. M., Grieshop, A. P., Lane, T. E., Pierce, J. R., and Pandis, S. N.: Rethinking organic aerosols: Semivolatile emissions and photochemical aging, Science, 315, 1259-1262, 10.1126/science.1133061, 2007.

Librando, V. and Tringali, G.: Atmospheric fate of OH initiated oxidation of terpenes. Reaction mechanism of $\alpha$-pinene degradation and secondary organic aerosol formation, J. Environ. Manage., 75, 275-282, 10.1016/j.jenvman.2005.01.001, 2005.

Kristensen, K., Cui, T., Zhang, H., Gold, A., Glasius, M., and Surratt, J. D.: Dimers in $\alpha$-pinene secondary organic aerosol: effect of hydroxyl radical, ozone, relative humidity and aerosol acidity, Atmos. Chem. Phys., 14, 4201-4218, 10.5194/acp-14-4201-2014, 2014.

Yasmeen, F., Vermeylen, R., Szmigielski, R., Iinuma, Y., Böge, O., Herrmann, H., Maenhaut, W., and Claeys, M.: Terpenylic acid and related compounds: precursors for dimers in secondary organic aerosol from the ozonolysis of $\alpha$- and $\beta$- pinene, Atmos. Chem. Phys., 10, 9383-9392, 10.5194/acp-10-9383-2010, 2010.

Aschmann, S. M., Reisseil, A., Atkinson, R., and Arey, J.: Products of the gas phase reactions of the OH radical with $\alpha$- and $\beta$-pinene in the presence of NO, J. Geophys. Res., 103, 25553-25561, 10.1029/98JD01676, 1998.

Gao, S., Surratt, J. D., Knipping, E. M., Edgerton, E. S., Shahgholi, M., and Seinfeld, J. H.: Characterization of polar organic components in fine aerosols in the southeastern United States: Identity, origin, and evolution, J. Geophys. Res., 111, D14314, 10.1029/2005JD006601, 2006.

Technical comments:
R2-6:) Line 46: Emerging work on DMS oxidation has found the formation of a key intermediate, hydroperoxymethyl thioformate (HPMTF). This intermediate is formed via an isomerization reaction and regenerates OH in the process. I would recommend adding context to this reaction and using it to understand the effect of DMS on the chamber observations.

A2-6:) Thank you for your valuable suggestion. We have added background on HPMTF in the introduction section. The following texts and figure were added in the new manuscript.

"During the daytime, DMS is consumed mainly by the reaction with OH, with H atom-abstraction reaction accounting for 65% (Berndt et al., 2019). A key branch point in DMS + OH is the methylthiomethylperoxy radical ($CH_3SCH_2OO$) formed from H-atom abstraction followed by $O_2$ addition. This subsequent reaction of $RO_2$ plays a dominant role in the product distribution of DMS. The $CH_3SCH_2OO$ radical can undergo bimolecular or unimolecular reactions (Jacob et al., 2024). Recent studies have identified a key intermediate, hydroperoxymethyl thioformate (HPMTF) (Ye et al., 2022; Veres et al., 2020). HPMTF is formed via secondary isomerization of the $CH_3SCH_2OO$ radical and regenerates the OH radical in the process (Berndt et al., 2019; Wu et al., 2015). In addition, DMS forms the major oxidation products, methanesulfonic acid (MSA), sulfuric acid ($H_2SO_4$), and sulfur dioxide via bimolecular pathways (with NO, $HO_2$ and $RO_2$) (Cala et al., 2023). The H-atom abstraction path of DMS is shown in Fig. 1. These products can contribute significantly to oceanic new particle generation, particle growth, and atmospheric chemical processes (Arquero et al., 2017; Fung et al., 2022)." (Page 2, Lines 50-59)

The scheme of the DMS isomerization channel is also presented in the new manuscript (Page 3, Lines 60-62).

[Figure]

Figure 1. H-abstraction reaction path of DMS with OH, the blue part indicates the intramolecular H-shift of $CH_3SCH_2O_2$ radical, and the red color indicates the formation of OH.

R2-7:) Line 70: How was the chamber run? Is this a batch mode or continuous flow method? Please explain in more detail how the chamber was run and what steps were taken to account for processes like dilution.

A2-7:) All experiments in the chamber were run in the batch mode. Therefore there was no dilution process. In addition, we have added some details of the experiments. The following texts were added in the new manuscript.

"All experiments in the chamber were run in batch mode." (Page 4, Line 85)

"Zero air generated by an air compressor combined with a zero-air generator (Model 1160; Thermo scisentific, USA) was used as the background gas and reactant carrier gas for the simulation experiments." (Page 4, Lines 90-92)

"Following this, hydrogen peroxide ($H_2O_2$, 30 wt%, Aladdin) was vaporized and flushed into the reactor to serve as a OH radical precursor. The concentration of $H_2O_2$ in all experiments was controlled at ~300 ppb. NO was introduced into the chamber from a gas cylinder (510 ppm in $N_2$, Qingdao Deyi Gas Company) using a mass flow controller." (Page 4, Lines 96-99)

"After all the reactants were introduced into the chamber, they were diluted to the desired volume by injecting zero air." (Page 4, Lines 99-100)

"All experiments were conducted at temperature of 299 ± 1 K and relative humidity (RH) of 30 – 40%." (Page 4, Lines 101-102)

R2-8:) Line 83: What are the concentrations of $H_2O_2$ used in the chamber? You use $H_2O_2$ photolysis to produce OH under high concentrations of NO. This will lead to a complex and high concentration mixture of $HO_2$, $RO_2$ and NO thus changing the fate of the peroxy radical formed in MT and DMS oxidation. Please devote more time to discussing this interaction.

A2-8:) The concentration of $H_2O_2$ in all experiments was controlled at ~300 ppb. The corresponding contents are added in the revised manuscript (Page 4, Lines 97-98).

Even though we have $H_2O_2$ as the OH precursor in our reaction systems, the simulation by MCM modeling revealed that the experiment still falls under the typical high $NO_x$ oxidation condition. **The details have been shown in the reply to Comment R2-2.**

R2-9:) Line 100: Wall loss for SOA and VOC can be an important driver of loss within an environmental chamber. Values are given for the wall loss terms without any validation or reasoning for the values. Could the authors please provide context and assumptions made for the values used.

A2-9:) The determination of wall losses of gases and particles is showed in the revised supplement (SI Pages 2-3, Lines 9-37; SI Pages 12-13, Lines 248-253).

(a) Gases

In order to quantify the gas wall loss, the target gas was injected into the clean reactor and its concentration over time was monitored. Wall deposition of gases inside the reactor can be considered as a first-order kinetic process as shown in equation (R12):

$$-\frac{d[C]}{dt} = -k_{ig,w}[C] \tag{R12}$$

$$\ln(\frac{[C]_t}{[C]_0}) = -k_{ig,w}t \tag{R13}$$

where [C] denotes the concentration of gas (ppb). $k_{ig,w}$ denotes the wall loss rate constant of gas (min⁻¹). Further integration of equation (1) leads to equation (R13). $[C]_t$ and $[C]_0$ denote the concentration of inorganic gases at 0 min and t min, respectively. We calculated the first-order wall loss rate constants for $\alpha$-pinene, DMS, NO, $NO_2$, $SO_2$, and $O_3$ using equation (2). The corresponding values were $3.159 \times 10^{-6}$, $8.982 \times 10^{-6}$, $1.178 \times 10^{-6}$, $1.241 \times 10^{-6}$, $2.878 \times 10^{-6}$, and $2.205 \times 10^{-6}$ min⁻¹ (Fig. R3). The calculated values are closer to the inorganic gases in the literature (Wu et al., 2007; Bloss et al., 2005; Metzger et al., 2008). Therefore, the wall losses for the gases are negligible.

(b) Particles

To perform the wall loss correction for the particles, the particles were injected the clean chamber. The number and mass concentrations of SOA produced in the formal experiment were corrected by measuring the wall loss constants of ammonium sulfate particles. Ammonium sulfate particles were generated as follows: ammonium sulfate solid particles were configured as a solution and small ammonium sulfate droplets were generated through an atomizer (Model 3076, TSI), which was then passed through a

silica-gel diffusion dryer and injected into the reactor. After the ammonium sulfate aerosols mixed well with zero air in the reactor, the number and mass concentrations of ammonium sulfate particles were monitored for 6-8 hours using SMPS. We assume that the particle wall loss is a first-order reaction and the particle wall loss rate constant, $k_{Dp}$, was defined in equation (R14) (Wang et al., 2018). $k_{Dp}$ is related to the particle size and time:

$$\ln[N_{Dp}(t)] = -k_{Dp}t + i \tag{R14}$$

where $N_{Dp}$ (nm) is the particle number concentration at size $D_P$ and time $t$, $k_{Dp}$ ($h^{-1}$) is the first-order particle wall-loss rate constant determined as the slope of the equation, and $i$ is an arbitrary constant. Then, for $k_{Dp}$ at a certain particle size, the four empirical parameters (a, b, c, d) are obtained by fitting with the four-parameter method as shown in equation (R15):

$$k_{Dp} = aD_p{}^b + c/D_p{}^d \tag{R15}$$

In this experiment, the parameters a, b, c and d were $1.95 \times 10^{-7}$, 1.72, 0.015 and 0.53, respectively. Therefore, the expression for the wall loss rate coefficient can be determined as $k_{Dp} = 1.95 \times 10^{-7} \times D_p{}^{1.72} + 0.015 \times D_p{}^{-0.53}$ (Fig. R4a). Then, the corrected SOA number concentration was obtained by equation (R16):

$$V_{Dp}{}^c(t) = V_{Dp}{}^m(t) + k_{Dp}\int_0^t V_{Dp}{}^m(t)dt \tag{R16}$$

Where $N_{Dp}{}^m(t)$ and $N_{Dp}{}^c(t)$ are measured and corrected particle volume concentration at size $D_P$. The particle mass concentration was corrected as above. As shown in Fig. R4b and d, the particle mass concentration and number concentration were well corrected.

[Figure]

Figure R3. Variation of gas concentration with time. (a) - (e) show the variation of VOC, NO, $NO_2$, $SO_2$ and $O_3$ with time, respectively.

[Figure]

Figure R4. Wall-loss correction of particle. (a) Variation of particle wall-loss coefficient with particle size. (b) Total particle mass concentration of aerosols generated from mixed system. (c) Size-dependent particle number concentration before correction from mixed system. (d) Size-dependent particle number concentration after correction from mixed system.

R2-10:) Figure 1: The axis's colors and labels do not match. I would recommend matching them to guide the readers eye.

A2-10:) Figure 1 (now Figure 2 in the current version) was revised in the manuscript (Page 8, Lines 185-187).

[Figure]

Figure 2. Variation of precursors with reaction time. Red and black dots indicate the results of smog chamber experiments and the curves indicate the results of MCM simulations. Blue dots indicate mass concentration of particles in smog chamber.

R2-11:) Line 150: Can you add a more in-depth analysis of DMS oxidation? You present one DMS chamber experiment and state that your observations don't match with Chen and Jang (2021). DMS has been studied through various oxidation methods and strategies. I would recommend further literature review to see if other work on DMS oxidation can match your observations and if not why. Just stating RH, oxidant and collection method does not explain the observed trends.

A2-11:) Previous studies have shown that the ratio of particulate MSA to non-sea salt sulfate varies between 0.05 and 0.75 and is usually below 0.5 (Bates et al., 1992; Ayers et al., 1996; Chen et al., 2012; Ayers et al., 1991). The ratio of MSA to $H_2SO_4$ in our study was ~0.67, which was consistent with the actual atmosphere. The multiphase chemical mechanism is complex, and the yields of $H_2SO_4$ and MSA depend on temperatures well as atmospheric composition (Mauldin Iii et al., 1999; Shen et al.,

2022). Moreover, field measurements of gas-phase MSA and $H_2SO_4$ show a wide range of concentrations.

In this study, we detected these two typical products of DMS using IC. However, the focus of our study is on the effect of these acidic products of DMS on $\alpha$-pinene SOA via heterogeneous reactions. In addition, DMS itself influences the generation of SOA in the mixed systems through OH regeneration. This aspect of the ratio of MSA to $H_2SO_4$ is less addressed in our study. To avoid a shift in focus, we have removed the relevant content in the revised manuscript.

If possible, we hope to deeply explore the formation of important products of DMS photooxidation in future studies. We also look forward to further understanding the effect of the ratio of MSA to $H_2SO_4$ on particulate matter formation.

R2-12:) Line 157: What is SOA in the DMS photooxidation? Could you please elaborate on what the components are of SOA that are not $H_2SO_4$. Could you please elaborate on what the new particles are in this case? Is the DMS SOA pure sulfuric acid clusters that other DMS derived species build upon. Do you have any indication of $NH_3$ or a gas-phase base to build with sulfuric acid.

A2-12:) Thank you for your valuable suggestion. DMS-SOA is defined as organosulfur compounds generated by DMS oxidation in the manuscript. Previous environmental monitoring data have shown that particulate matter generated by DMS photooxidation is mainly composed of sulfuric acid and organosulfur compounds. Organosulfur compounds such as DMSO, MSIA, and MSA, as well as other products, are also produced from DMS oxidation via the OH-addition pathway. Many of these products partition into the condensed phase, and extensive data sets exist for methanesulfonate ($CH_3SO_3^-$, MS) and non-sea-salt sulfate (nss-$SO_4^{2-}$), the deprotonated forms of MSA and $H_2SO_4$, respectively (Barnes et al., 2006). Methylsulfonylperoxynitrate (MSPN, $CH_3SO_2OONO_2$) have been observed as products of the OH-radical-initiated oxidation of DMS in laboratory studies (Ye et al., 2021).

In fact, we also attempted to utilize UPLC/ESI-Q-TOFMS to detect the particle-phase product of DMS, i.e., Exp. D-2. Unfortunately, we did not detect any valuable signals or substances. We suspect that the insensitivity of the extraction process and program settings of the offline technique to the sulfur-containing products of DMS is the cause. Our exploration of the composition of SOA focuses on the effect of DMS on $\alpha$-pinene SOA, while assuming the particulate phase products of DMS to be typical final products such as $H_2SO_4$ and MSA. In addition, Chen and Jang (2012) showed that the contribution of detectable $H_2SO_4$ and MSA, etc., to the offline analysis of the particulate phase products of DMS is not exactly equal to the mass concentration of OM, which is similar to our results. Currently, researchers typically use on-line instruments (e.g., AMS) to detect the particulate phase products of DMS (Ye et al., 2022; Ye et al., 2021). This makes the analysis of the products of DMS more sensitive. We expect that in future studies, the gas-phase and particle-phase components of DMS can be investigated in more detail at the molecular level using AMS, CIMS, etc.

Regarding the form of $SO_4^{2-}$ in the particles, we found that previous studies did not mention the effect of $NH_3$ on $SO_4^{2-}$, whether $SO_2$ or DMS was added as a reactant (Ye et al., 2018; Liu et al., 2019; Lewandowski et al., 2015; Liu et al., 2017). In our experiments, we detected the amount of $NH_4^+$ in the particulate phase using IC to be between 0.1 and 0.5 μg m$^{-3}$ in the presence of DMS. The concentration of $NH_4^+$ was 1 to 3 orders of magnitude lower than the mass of the other products of DMS. Alternatively, very small amounts of ammonium sulfate clusters do not affect the mass concentration of SOA. Therefore, we have neglected the effect of $NH_4^+$ on $SO_4^{2-}$. We concluded that all $SO_4^{2-}$ quantified by ion chromatography is $H_2SO_4$ in our experiments.

We attribute particle nucleation to two aspects.

$H_2SO_4$ affects the formation of particulate matter (Vivanco et al., 2013; Liu et al., 2016). The role of $H_2SO_4$ in new particle formation has been well studied in previous studies (Berndt et al., 2005; Zhang et al., 2012; Sipilä et al., 2010; Kirkby et al., 2011; Almeida et al., 2013). $H_2SO_4$ promotes the formation of new particles by participating in nucleation or forming organosulfates. $H_2SO_4$ increases particle mass concentration and LVOC production via heterogeneous reactions (Xu et al., 2021; Zhang et al., 2023). We detected a certain amount of $SO_4^{2-}$ using IC, which is attributed to $H_2SO_4$.

LVOCs/ELVOCs formed during reactions were previously found to be involved in the new particle formation (Wildt et al., 2014; Kirkby et al., 2016) and growth of particles, such as highly oxidized multifunctional molecules (HOMs), dimers, and trimers, etc. (Ehn et al., 2014; Kirkby et al., 2016). HOMs of $C_8$ - $C_{10}$ were identified in our study. These molecules are important for new particle generation.

The corresponding contents are added in the manuscript.

"Previous environmental monitoring data have shown that particulate matter generated by DMS photooxidation is mainly composed of $H_2SO_4$ and organosulfur compounds (Gaston et al., 2010; Chen and Jang, 2012; Veres et al., 2020). Organosulfur compounds such as DMSO and MSA, as well as other products, are also produced from DMS oxidation via the OH-addition pathway. Many of these products partition into the condensed phase (Barnes et al., 2006). We attempted to utilize UPLC/ESI-Q-TOFMS to detect other particle-phase products of DMS. Unfortunately, we did not detect any valuable signals or substances. We suspect that the insensitivity of the extraction process and program settings of the offline technique to the sulfur-containing products of DMS is the cause." (Page 8, Lines 178-184)

"The particulate matter generated by DMS photooxidation mainly contains two types of components, $H_2SO_4$ and SOA (or organosulfur compounds)." (Page 8, Lines 188-189)

"[b] IC detection, particle-phase products generated by DMS photooxidation. $NH_4^+$ was hardly detected. All $SO_4^{2-}$ were detected by IC as $H_2SO_4$." (Page 9, Line 198)

R2-13:) Table 2: The connection between max $O_3$, $SO_2$, and $NO_x$ and max SOA is not a straight forward concept that should be evaluated within the text. Presenting one or two of the time traces could be an informative way to understand when the peaks are occurring and how that relates to the steady state or end of experiment concentration. The amount of $O_3$ produced across the experiments varies by a decent amount. I would

recommend addressing this variability and seeing if its formation can help understand what is happening within the chamber. Also is O$_3$ + MT important at these concentrations?

A2-13:) Thank you for your valuable suggestion. We discussed the relationship between inorganic gases and SOA generation in our experiments in Sec. S5 of the original supplement. The trend of the inorganic gas over time was shown.

We found that the O$_3$ generated in our experiments affect little on the oxidation of $\alpha$-pinene and the conversion path of the intermediates by comparing the reaction rates of O$_3$ and OH with $\alpha$-pinene.

We calculated the rates of reaction of different systems of $\alpha$-pinene with O$_3$ and OH, respectively. The corresponding contents are added in the revised supplement (SI Page 11, Lines 236-246; SI Page 18, Lines 276-277). The rate constants of O$_3$ + $\alpha$-pinene and OH + $\alpha$-pinene reactions were determined to be $8.7 \times 10^{-17}$ and $5.4 \times 10^{-11}$ cm$^3$ molecule$^{-1}$ s$^{-1}$, respectively (Zhang et al., 1992). The reaction rate is calculated as equations (R17) and (R18):

$$v_{O3+\alpha\text{-pinene}} = k_{O3+\alpha\text{-pinene}}[\alpha\text{-pinene}] \tag{R17}$$

$$v_{OH+\alpha\text{-pinene}} = k_{OH+\alpha\text{-pinene}}[\alpha\text{-pinene}] \tag{R18}$$

where $v_{O3+\alpha\text{-pinene}}$ and $v_{OH+\alpha\text{-pinene}}$ denote the reaction rate (s$^{-1}$) of $\alpha$-pinene with O$_3$ and OH, respectively. $k_{O3+\alpha\text{-pinene}}$ and $k_{OH+\alpha\text{-pinene}}$ denote the rate constants (cm$^3$ molecule$^{-1}$ s$^{-1}$) of the reaction of $\alpha$-pinene with O$_3$ and OH, respectively. [$\alpha$-pinene] and [DMS] denote the measured concentrations of the two VOCs, respectively (cm$^3$ molecule$^{-1}$).

Figure R5 below demonstrates the trend of the reaction rate with time for different oxidation systems. The rate of reaction between $\alpha$-pinene and OH has been much greater than that between $\alpha$-pinene and O$_3$ during the time period when $\alpha$-pinene was present. The reaction of O$_3$ with $\alpha$-pinene was consistently close to 0. This suggests that O$_3$ has a negligible effect on the consumption of $\alpha$-pinene.

Hence, the $\alpha$-pinene reaction with O$_3$ would be less important than the reaction with OH in our study.

[Figure]

Figure R5. The reaction rate of $\alpha$-pinene with $O_3$ or OH in different experiments.

Inorganic gases are described as shown below. The corresponding contents can be found in the revised supplement (SI Pages 5-6, Lines 81-88, 103-106, SI Page 14, Lines 254-255).

Figure S3 shows the reaction profiles of inorganic gases (i.e., NO, $NO_x$, $O_3$, and $SO_2$) over the course of the chamber experiments. NO was consumed more rapidly in the mixed experiments than in Exp. A-1. This is probably due to the higher concentration of $RO_2$ in the mixed experiments since the oxidation of both DMS and $\alpha$-pinene produces $RO_2$ that can react with NO. In addition, it is found that the $SO_2$ concentration increases with the increasing initial concentration of DMS in the mixed system, which is due to the $SO_2$ production from DMS photooxidation. However, there is no significant difference in the maximum ozone concentration with increasing DMS, indicating the weak effect of DMS on $O_3$ production (Chen et al., 2019).

Figure 2 also shows the evolution of the particle mass concentration after the particle wall loss correction. It is demonstrated that the particle mass start to increase before $\alpha$-pinene has been fully reacted, which is consistent with previous studies (Kari et al., 2017). In addition, the SOA generation occurred after NO is consumed to ~0 ppb due to suppression of hydroperoxide formation by the $RO_2$ + NO reaction (Liu et al., 2022).

[Figure]

Figure S3. Time profiles of inorganic gases (i.e., NO, NO$_x$, O$_3$, and SO$_2$)

[Figure]

Figure 2. Variation of precursors with reaction time. Red and black dots indicate the results of smog chamber experiments and the curves indicate the results of MCM simulations. Blue dots indicate mass concentration of particles in smog chamber.

We have also added a description of the relevant details in the revised manuscript.

"The time series of inorganic gases and the related presentation of the connection with SOA formation are also presented in Sect. S5 of the supplement." (Page 9, Lines 196-197)

"We calculated the reaction rates of $O_3$ or OH with $\alpha$-pinene, respectively. It is shown that the effect of $O_3$ was very low (Fig. S11). Therefore, the effect of $O_3$ on the $\alpha$-pinene SOA is ignored in the discussion. Details can be found in Sect. S10 in the supplement." (Page 11, Lines 235-237)

R2-14:) Table 3: The header of Table 2 is added to the start of Table 3. Please fix this.

A2-14:) We have corrected the header of this table.
Page 9, Line 198 in the revised manuscript.

Table 2. Experimental results of particle-phase components in photooxidation of DMS/$\alpha$-pinene/NO$_x$ systems.

| Exp. No. | [Total particles] [a] $\mu g\ m^{-3}$ | [H$_2$SO$_4$][b] $\mu g\ m^{-3}$ | [MSA][b] $\mu g\ m^{-3}$ | [SOA$_m$][c] $\mu g\ m^{-3}$ | Y$_m$[d] | [SOA$_p$][e] $\mu g\ m^{-3}$ |
|---|---|---|---|---|---|---|
| individual $\alpha$-pinene | | | | | | |
| A-1 | 269.5 | - | - | 269.5 | 0.16±0.02 | |
| individual DMS | | | | | | |
| D-1 | 177.2 | 50.8 | 32.83 | 116.2 | 0.25±0.03 | |
| mix $\alpha$-pinene and DMS | | | | | | |
| AD-1 | 296.3 | 15.0 | 22.2 | 270.8 | 0.14±0.02 | 216.7 |
| AD-2 | 422.3 | 15.7 | 22.5 | 400.1 | 0.20±0.02 | 270.3 |
| AD-3 | 572.6 | 45.4 | 24.9 | 507.5 | 0.24±0.02 | 425.8 |
| AD-4 | 714.4 | 55.4 | 50.5 | 607.7 | 0.25±0.03 | 648.7 |
| AD-5 | 683.0 | 48.8 | 36.2 | 613.1 | 0.24±0.02 | 708.2 |
| AD-6 | 551.5 | 35.3 | 8.5 | 504.2 | 0.19±0.02 | 680.9 |
| AD-7 | 539.9 | 48.4 | 16.8 | 476.0 | 0.19±0.02 | 677.5 |
| AD-8 | 364.4 | 68.7 | 0.1 | 237.1 | 0.08±0.01 | 537.6 |
| AD-9 | 289.9 | 83.2 | 7.0 | 154.0 | 0.06±0.01 | 436.5 |

[a] The mass concentration of particles generated by SMPS, corrected for particle wall loss, was calculated as a particle density of 1.2 g cm$^{-3}$.
[b] IC detection, particle-phase products generated by DMS photooxidation. NH$_4^+$ was hardly detected. All SO$_4^{2-}$ were detected by IC as H$_2$SO$_4$.
[c] The measured SOA mass concentration is expressed as [Total particles]$_{after-correction}$ × (1 - [H$_2$SO$_4$] / [Total particles]$_{before-correction}$).
[d] [SOA$_m$] / ($\Delta$[$\alpha$-pinene] + $\Delta$[DMS]), as mixed yield. Error bars indicate SMPS instrument error of 10%.
[e] The predicted SOA mass concentration by using mass-dependent SOA yields of $\alpha$-pinene and DMS.

R2-15:) Line 205: Needs citation for the isomerization rate. Additionally, I would recommend reviewing the literature around HPMTF formation as new slower rates exist compared to Wu et al. It is also important to caution the 43% increase in OH with

comments about how much isomerization is occurring in the chamber work presented here. High NO, HO2 and RO2 likely present here could arrest this channel.

A2-15:) Thank you for your valuable suggestion. We reviewed the literature on HPMTF and summarized the isomerization rate of $CH_3SCH_2O_2$ radical in Table R2 below. The temperature range of our experimental setup was $299 \pm 1$ K, similar to those of (Jacob et al., 2024; Assaf et al., 2023)'s studies. Therefore, we set the isomerization rate of $CH_3SCH_2O_2$ radical to 0.06 s$^{-1}$. We cite the isomerization rate in the relevant part of the new manuscript.

Table R2. Literature summary of isomerization rates of $CH_3SCH_2O_2$ radicals.

| | Rate of isomerization (s$^{-1}$) | T (K) | Ref. |
|---|---|---|---|
| 1 | 2.1 | 293 | (Wu et al., 2015) |
| 2 | $0.23 \pm 0.12$ | $295 \pm 2$ | (Berndt et al., 2019) |
| 3 | 0.2 | 293 | (De Jonge et al., 2021) |
| 4 | 0.041 | 293 | (Veres et al., 2020) |
| 5 | $0.06 \pm 0.02$ | 298 | (Jacob et al., 2024; Assaf et al., 2023) |

We reviewed the Berndt et al. (2019)' study. Their study showed that a 42% increase in OH occurs when the percentage of isomerization channel exceeds 95%. We revised this in the new manuscript.

We fitted the time series of $HO_2$, $RO_2$, NO, and $CH_3SCH_2O_2$ using the modified MCM model. We have added reactions related to the isomerization pathway of the $CH_3SCH_2O_2$ radical to the MCM model. We calculated the rate for each reaction path and evaluated the absolute amount of the isomerization channel of the $CH_3SCH_2O_2$ radical using the MCM model, **see our reply to Comment R2-3 for details.**

Related results are displayed in Fig. R6 and R7 below. As can be seen from Fig. R6, the MCM model results demonstrate the importance of the isomerization channel rate. The rate at which isomerization occurs for the oxidation of DMS alone does not change significantly and rises only slowly after 200 min. However, most of the mixed experiments showed an increase in the slope of the rate of isomerization channel after a reaction time of ~75 min. The isomerization was second important channel that only slower than the NO + $CH_3SCH_2O_2$ reaction. Relative percentage of $HO_2$, $RO_2$, NO and isomerization channel of $CH_3SCH_2O_2$ radical further demonstrates this (Fig. R7). Among the four reaction channels, the isomerization pathway of $CH_3SCH_2O_2$ radical accounts for a sufficient percentage to compete with the other bimolecular reaction channels. Therefore, it suggests that isomerization is an important reaction channel of $CH_3SCH_2O_2$ radical, and is not arrested by the high NO, $HO_2$ and $RO_2$ concentrations.

[Figure]

Figure R6. Rates of each reaction channel of the $CH_3SCH_2O_2$ radical calculated by the MCM model in DMS individual and mixed experiments.

[Figure]

Figure R7. Relative percentage of $HO_2$, $RO_2$, NO and isomerization channel of $CH_3SCH_2O_2$ radical obtained from MCM model fitting in different mixing experiments.

The corresponding contents are added in the revised supplement (SI Pages 7-8, Lines 128-141; SI Page 19, Lines 278-280) and manuscript (Page 13, Lines 273-277).

We have also added relevant details about the isomerization channel of DMS in the revised manuscript. As follows:

"As shown in Fig.1, DMS forms $CH_3SCH_2O_2$ radical, which undergoes an isomerization process to form $HOOCH_2SCHO$ product, accompanied by OH regeneration (Berndt et al., 2019; Wu et al., 2015). The isomerization rate of the $CH_3SCH_2O_2$ radical is 0.06 s$^{-1}$ (Assaf et al., 2023; Jacob et al., 2024)." (Page 12, Lines 259-261)

"Their results of atmospheric chemistry simulations demonstrate the predominance ($\geq$ 95%) of $CH_3SCH_2O_2$ isomerization. We fitted the contribution of the different reaction channels of the $CH_3SCH_2O_2$ radical using the MCM model (Fig. 5a). The calculations are detailed in Sec. S7. We find that the isomerization channel is a major reaction channel in our experiments following the NO + $CH_3SCH_2O_2$ channel. (Fig. S12)." (Page 12, Lines 263-267)

R2-16:) Figure 6: "a" and "b" are not labeled on the Figure.

A2-16:) "a" and "b" have been labeled on the Figure in the revised manuscript (Page 16, Lines 341-343).

R2-17:) Figure 7: Please define and describe the meaning of "low, middle, and High" for the mixtures within the description.

A2-17:) "Low, medium and high" represent $\Delta[DMS]/\Delta[\alpha\text{-pinene}]$ in mixed experiments, indicating different mixing ratios. Specifically, "low" represents Exp. AD-10 below the turning point, "medium" represents Exp. AD-11 at the turning point, and "high" represents Exp. AD-12 below the turning point. The corresponding contents are added in the manuscript (Page 17, Lines 363-365).

R2-18:) Figure 8: A hydrogen shift mechanism across 4 carbons is presented and assumed to be a reaction within the chamber. I am not aware of this being a known reaction. Could the authors please provide evidence or citations that would support this mechanism. I understand that the bicyclic nature of the molecule could bring the hydrogen and alkyl radical close, but replacing a secondary radical with a primary seems highly unlikely.

A2-18:) Thank you for your valuable suggestion. We have corrected the mechanisms. We think that the H-shift path shown in Fig. 8 from the original manuscript indeed cannot occur after reviewing the literature. We have removed the second pathway in Fig. 8 from the original manuscript after revision. At the same time, we have modified the first pathway to other pathways, while proposing $C_{10}H_{15}NO_6$ to be the other isomer. The molecular structure of the modified $C_{10}H_{15}NO_6$ is a ring-opening product, which is

oxidized from pinonaldehyde (Eddingsaas et al., 2012a). In addition, we retained the molecular structure of the dimer $C_{20}H_{33}NO_8$, which is derived from Draper et al. (2015). The following texts and figures were added in the revised manuscript.

"Figure 12b shows the possible pathway of ON formation. In the presence of $NO_2$, the hydrogen atoms on the carbon chain of the typical product pinonaldehyde can be readily oxidized to form nitrogen-containing carboxylate products by the addition of oxygen, i.e., $C_{10}H_{15}NO_6$ (MW 245) (Boyd et al., 2015; Kim et al., 2012; Eddingsaas et al., 2012b)." (Page 20, Lines 417-420)

"In addition, Fig. 12b demonstrates the possible structure of a high molecular weight oligomer generated in the individual $\alpha$-pinene experiments: $C_{20}H_{33}NO_8$ (MW 415). It is speculated that $RO_2$ tends more towards isomerization processes such as autoxidation compared to fragmentation reaction (Draper et al., 2015). This pathway increases the possibility of oligomerization of $RO_2 + RO_2$ and $RO_2 + HO_2$ in individual $\alpha$-pinene oxidation." (Page 20, Lines 423-426)

The mechanisms related to ONs have also been modified, as shown below (Page 21, Lines 427-430).

[Figure]

Figure 12. Proposed formation mechanisms and structural for organosulfate (a) and organic nitrates (b) in SOA. Red, blue and black in the boxes refer to the products identified by $\alpha$-pinene-only SOA products, mixed-only SOA products and $\alpha$-pinene-mixed-both SOA products, respectively.